

# Lowlands fluvial sedimentation enlightens glacial dynamics in narrow valleys during the Last Glacial Maximum (Venetian Forealps, Italy)

Sandro Rossato[1], Anna Carraro[2], Giovanni Monegato[2], Paolo Mozzi[1], Fabio Tateo[2]

[1]Department of Geosciences, University of Padova, 35131, Italy

[2]Institute of Geosciences and Earth Resources (IGG) – National Research Council (CNR), Padova, 35131, Italy

*Correspondence to*: Sandro Rossato (sandro.rossato@unipd.it)

**Abstract.** During the Last Glacial Maximum (LGM), most of the major glaciated basins of the European Southern Alps had piedmont lobes with large outwash plains; only few glaciers remained within the valley. The formers left well-preserved terminal moraines, whose investigation allowed to infer their evolution and chronology. Valley glaciers remnants, on the

contrary, are often scantly preserved and changes can be detected only through the correlation with the glaciofluvial deposits in downstream alluvial basins. The Brenta glacial systems dynamics in its terminal tract was inferred through a wide range of sediment analysis techniques on an alluvial stratigraphic record of the Brenta megafan (NE Italy) and the mapping of in-valley glacial/glaciofluvial remnants. Glaciers flowing across narrow gorges turned out to be possibly slowed/blocked by such morphology and, if a lateral valley exists, glacial/sediment fluxes can be diverted. Moreover, narrow valleys may induce

glaciers to bulge and form icefalls at their front, preventing the formation of terminal moraines. The Brenta glacier was probably slowed/blocked by the narrow Valsugana gorge downstream of Primolano and was effectively diverted eastwards across a windgap (Canal La Menor valley), joining the Cismon/Piave glaciers near Rocca and ending ~2 km downstream. The Cismon River started to flow along its present path just before 27 ka cal BP, while the Piave catchment contributed to the Brenta system at the acme of LGM, from ~27 to, at least, ~19.5 ka cal BP.

Our investigation shows that glacial catchments may significantly vary over time during a single glaciation in rugged Alpine terrains. Sand petrography and chemical/mineralogical composition of sediments are good tracer of such variations, that reflects in the glacial and glaciofluvial systems and can be recognized in the alluvial stratigraphic record far downstream from the glacier front.

## 1 Introduction

Mountain glaciers are complex systems, whose evolution affects both their mountain basins and the alluvial plains that receive the glaciofluvial water-and-sediment flux (e.g. Russell et al., 2006). Few minor valley glaciers are currently present in the highest areas of the European Alps (Evans, 2006), whereas during Pleistocene glaciations large ice-streams flowed along most of the Alpine valleys, leading to deep landscape modifications (e.g. Koppes and Montgomery, 2009; Preusser et al., 2010; Wirsig et al., 2016). Most of the major glaciated basins had piedmont lobes with large outwash plains, where the stratigraphic



reconstruction and available chronology allowed to infer the evolution of the glacial system (Monegato et al., 2007; Preusser et al., 2011; Ravazzi et al., 2012; Fontana et al., 2014). However, there are catchments whose glacier remained within the valley and their evolution could be detected through the correlation with the related glaciofluvial deposition in the piedmont area (van Husen and Reitner, 2011; Rossato et al., 2013).

The Last Glacial Maximum (LGM) was the last cold extreme on Earth and it provided the best-preserved sedimentary and geomorphic record among all Pleistocene glaciations (Bowen, 2009; Clark et al., 2009; Hughes and Gibbard, 2015). In Europe, the Alps and their foreland constitute a key region for LGM studies, as evidence of this event is widespread and chronologically well framed between 30 and 17.5 ka cal BP (e.g., Ivy-Ochs et al., 2008; Ivy-Ochs, 2015; Rossato and Mozzi, 2016; Monegato et al., 2017). Moreover, they constitute an effective barrier for wind circulation (Florineth and Schlüchter, 2000; Luetscher et

al., 2015) and offer the unique opportunity to validate climate models (e.g. Smiatek et al., 2009; Torma et al., 2015) and study how glacial processes affect mountain chains (e.g. Norton et al., 2010). LGM glaciers were filling the valleys fed by the major accumulation zones located in the axial and highest part of the Alps (e.g., van Husen, 1987; Kelly et al., 2004; Wirsig et al., 2016). In the southern alpine sector, due to different wind circulation regime (Luetscher et al., 2015), topography greatly influenced glacier evolution (Wirsig et al., 2016). Alpine glaciers, and their sedimentary outputs, proved to react differently to

climatic signals, both along the North-South (e.g. Luetscher et al., 2015) and the West-East directions (Becker et al., 2016; Monegato et al., 2017; Seguinot et al., 2017). Differences arise also when dealing with neighboring glacial systems, due to their size, catchment topography and possible glaciers confluences and transfluences (e.g., Kelly et al., 2004; Monegato et al., 2007; Rossato et al., 2013).

The present availability of a wide range of proxies, coupled with the increased accuracy of geochronological methods (Brauer

et al., 2014), allow to a better assessment of LGM glaciers' behavior compared to earlier studies in the Alps, mostly based on landforms and deposits characterization (e.g. Penck and Brückner, 1909; Sacco, 1937; Venzo et al., 1977; Schlüchter, 1986; van Husen, 1987). The analysis of loess and palaeosols has been coupled with biological proxies, such as pollen, chironomids, charcoal and many more (Heiri et al., 2014; Samartin et al., 2016). Petrographic/mineralogical study of sediments have supported palaeoenvironmental reconstructions and allowed to infer variations in the sedimentary systems (Garzanti et al.,

2011).

In this paper we investigate the interaction between the LGM glaciers in the middle Brenta valley (also known as "Valsugana", Italian Forealps) and the related glaciofluvial system in the piedmont plain (Fig. 1). During the LGM, the Brenta glacial system received contribution from the major Adige (Etsch) glacier (Trevisan, 1939; Tessari, 1973; Avanzini et al., 2010; Rossato et al., 2013), and fed a fluvial megafan, which is one of the most prominent sedimentary alluvial systems on the southern side of

the Alpine chain (Mozzi, 2005; Fontana et al., 2014). The main aim of this paper is to explore and define the possibility of correlating glacial advances and transfluences in mountain areas with sedimentation pulses in the lowlands using integrated geomorphic, sedimentary, petrographic, mineralogical, geochemical and geochronological evidences.



## 2 Setting

The Venetian Forealps are geologically detached from the Dolomites by the Valsugana fault (Castellarin et al., 2006) and are characterized by a belt of carbonate plateaus deeply carved by the lower reaches of the Astico, Brenta, Cismon and Piave valleys (Fig. 1). Except for the Astico River, whose catchment is restricted to the western part of the Venetian Forealps, the others have their upper catchments in the Dolomites, which include the Permo-Triassic sedimentary successions (sandstones, dolostones, limestones and volcanic rocks), the low-grade crystalline basement and the Permian porphyries (e.g., Bartolomei et al., 1969; Avanzini et al., 2010; Fig. 1). The Brenta and Cismon catchments also include the Permian plutonic rocks of Cima d'Asta (Fig. 1). In the footwall of the Valsugana Fault the Jurassic-Tertiary sedimentary succession crops out, including different types of limestones, turbiditic sequences and the terrigenous units of the Southalpine foredeep (e.g., Massari et al., 1986; Barbieri and Grandesso, 2007; Stefani et al., 2007). In particular, in the Cismon catchment, all around Lamon (Fig. 1), micritic limestones crop out extensively (Tessari, 1939). Here, a remarkable fluvial aggradation took place during/after deglaciation, later followed by river incision and a five-step terrace staircase is preserved, the highest terrace being located at about 600 m a.s.l., 200 m above the Cismon valley bottom. The deposition of such an amount of sediment was due probably to the damming of the Cismon valley south of Lamon, possibly by a large dead-ice mass or a landslide event, but no direct evidence of it has been found (Tessari, 1939).

The plateaus characterizing the Venetian Forealps (Castiglioni et al., 1988) have mean elevations from 900 to 2000 m a.s.l. and they hosted small cirque or plateau glaciers during the Last glaciation (Trevisan, 1939; Carraro and Sauro, 1979; Baratto et al., 2003; Barbieri and Grandesso, 2007). The major Astico (Cucato, 2001; Rossato et al., 2013) and Piave (Venzo, 1977; Carton et al., 2009) valleys were carved by ice-streams that reached the lower valley segments and left well-preserved terminal moraines. Despite the large megafan ascribed to the LGM (Mozzi, 2005; Fontana et al., 2008; Rossato and Mozzi, 2016), the Brenta glacier is the only major system in the south-eastern Alps that left no clear evidence of terminal moraines (Castiglioni, 2004). The reason why the Brenta system did not follow such evolutionary path remains unresolved, although some early authors already dealt with this challenging topic (e.g. Taramelli, 1882; Penck and Brückner, 1909; Castiglioni, 1940). Taramelli (1882) and Castiglioni (1940) placed the Brenta glacier front near the town of Valstagna, about 10.5 km south of the confluence between the Brenta and Cismon valleys (Fig. 1), while Penck and Brückner (1909) located it about 6 km further to the south, near the town of Solagna. Due to the lack of direct geomorphic or sedimentary evidence at the glacier's front, both interpretations were based on the speculation that the glacier thinned gradually downstream of the last preserved lateral moraine at Enego (elevation 790-760 m a.s.l.). The terminal tract of the Cismon valley, named Corlo valley, is closed by a dam and filled by the artificial lake of the same name (Fig. 1) since 1953 and any direct observation of sediments is prevented.

The Brenta fluvial megafan is part of the foreland basin of the uplifting Eastern Southalpine chain (e.g., Castellarin et al., 2006; Stefani et al., 2007). This landform extends from the Brenta valley mouth to the Venetian coastline. It can be roughly separated into two sectors: i) piedmont (~20 km from the apex), made of gravel and with an average topographic gradient of 3-5‰, ii)



low plain, made of sandy fluvial ridges and silty-clayey floodplains, with a decreasing of topographic gradient to less than 1‰ (Mozzi, 2005; Fontana et al., 2008; Mozzi et al., 2013).

## 3 Methods

This investigation was carried out through field survey, coring, petrographic and mineralogical analyses and remote sensing.
All methods and results are presented independently, whilst their meaning, implications and interconnections are discussed thereafter.

### 3.1 Field survey

An extensive mapping of the sediments cropping out in the area of the middle Valsugana and the junctions with Canal La Menor and lower Cismon (Corlo) valley, was performed (Fig. 2). Sedimentary facies were observed and described in the
outcrops to identify their original depositional setting; particular attention was devoted to the lithology of the clasts, which provided information on the provenance of the sediments. As it is possible to appreciate in Fig. 1, the geology of the eastern Southern Alps allows a clear distinction of each drainage basin with different petrographic signatures. Granites, porphyry and metamorphic rocks are indicators of Brenta and Cismon valleys and, thanks also to their high resistance to weathering, can help in identifying glacial deposits.
Relevant landforms were accurately surveyed, with particular attention to the LGM moraines which are generally well preserved and quite easily identifiable in these areas, where little anthropogenic activity took place.

### 3.2 Cores

Two 30-m-long cores were drilled near the city of Piazzola sul Brenta, in the upper part of the Brenta megafan, about five km apart and at a topographic elevation of about 30 m a.s.l. (Fig. 3). These cores were part of a pilot study on groundwater
geochemistry (Carraro et al., 2013; 2015). They were described basing on the lithofacies of the sediments and sampled for sand petrography and mineralogical analyses. Five organic samples were collected from the inner part of the cores to minimize contamination and dated with the [14]C AMS method at the radiocarbon laboratory of the University of Zurich (Switzerland) (Table 1). OxCal software (version 4.2, Bronk Ramsey, 2009; IntCal13 calibration curve, Reimer et al., 2013), has been used to calibrate laboratory ages.
A single hand-core was drilled in the Guarda area, on the northern flank of the Canal La Menor valley, uphill of an elongated ridge that has been identified as a LGM lateral moraine (see Results chapter and Fig. 2 for location), at about 660 m a.s.l.. This borehole has been realized with a hand-auger (Edelman combination type, Ejikelkamp[TM]); this equipment allows to obtain a semi-disturbed sequence of fine sediments, with the limitation that the maximum grain size of the sediments that can be sampled is coarse sand to fine gravel.



### 3.3 Sand petrography

Ten samples were collected from the two long cores for provenance analysis. The sandy fraction (0.0625÷2 mm) was isolated and impregnated in an epoxy resin according to Gazzi et al. (1973) methodology, in order to obtain samples for thin-section analysis. These were subsequently stained with alizarine-red solution for the determination of the carbonate phases. Gazzi-Dickinson procedures (Ingersoll et al., 1984) was adopted for counting and 400 points were checked, using a 0.5 mm grid spacing, for each section. Data and parameters were reported in Table 2, plotted in ternary diagrams. Petrofacies of modern rivers (Garzanti et al., 2006; Monegato et al., 2010) and local late-Pleistocene deposits (Monegato et al., 2011; Rossato et al., 2013) were considered for comparison to assess the provenance of the different stratigraphic intervals. The statistical spreadsheet by Vezzoli and Garzanti (2009) was adopted to assess the contribution of each river system.

### 3.4 X-ray diffraction (XRD)

In order to minimize preferred orientation, samples were pulverized and backloaded. A Philips X'Pert Pro diffractometer (Cu tube and secondary monochromator) was adopted. The mineral constituents were quantified using HighScore plus software following Rietveld refinement (Young, 1993); zincite (ZnO) was added as an internal standard (sample:standard = 10:1 weight ratio). It allows also to estimate the amorphous components (labelled Am.XRD, in Stab. 1).

To check the reliability of mineral quantification, the chemical composition of samples was calculated using stoichiometric mineral composition (for quartz, calcite, dolomite, feldspars, illite and kaolinite); a Fe-rich, Mg-poor dolomite was also included in the calculation (according to the XRD evidences of non-stoichiometric dolomite in almost all the samples), two Al-chlorites (Fe- or Mg-enriched) and organic matter. The comparison between calculated and measured chemical composition was considered satisfactory (total least square within 10 for 29 samples over 32).

### 3.5 Chemical composition

The major elements, some trace elements (including N) and the organic carbon ($C_{org}$) were determined as reported by Carraro et al. (2015). Sample were digested with concentrated $HNO_3$ + HCl + HF at 140 °C for 90 minutes, then rinsed with $H_3BO_3$ and heated at 140 °C for 60 minutes. Afterwards, the solutions were analyzed by inductively coupled plasma (ICP-OES) and atomic absorption spectroscopy (AAS). The loss on ignition (LOI) was measured after heating the sample to 860 °C for 20 minutes and to 980 °C for 2 hours.

An automatic elemental analyzer was used to obtain $C_{org}$ and total N for the carbonate-free residues. The measures were repeated at least twice using Ag sample containers.

All the mineralogical and chemical data were used for hierarchical cluster analysis (SPSS), in order to point out sample similarities and/or anomalies. Five dark sediments were grouped in the same cluster; this group is characterized by a percentage of organic matter higher than 12 %. This threshold value was used for peat identification (see Carraro et al., 2015 for analytical

details). Macroscopic features (black color and fine-grained) were used for peat identification when chemical data were not available.

## 3.6 Remote sensing

Aerial photographs and satellite images of public use (Google Earth and Bing databases) were processed and analyzed for the
mapping of relict landforms. Panoramic and detail photos were acquired during field surveys and subsequently used in conjunction with the other images.

A Digital Elevation Model (DEM) provided by the Veneto Region, based on topographic data derived from 1:5000 topographic maps (5 m cell-size; XY accuracy: 2 m - http://idt.regione.veneto.it/app/metacatalog/) was used. It covers the entire study area, assuring a uniform accuracy of investigation. The various DEM tiles were assembled with ArcGis (10.4.1 version) to better
process them and to obtain a uniform visualization. All topographic profiles here presented were based on this DEM.

## 4 Results

### 4.1 Field survey

The field survey allowed to identify some key stratigraphic sections and outcrops that were regarded to be representative of specific sectors of the study area (see Fig. 2 for location). Their descriptions are presented here in geographical order, from
East to West and from North to South.

### 4.1.1 "Seren valley" and "Roncon" outcrops

In the Seren valley bottom (390 m a.s.l.), widespread outcrops of matrix-supported diamicton can be found (Fig. 2). These deposits are characterized by centimetric clasts with sparse boulders, sub-rounded to angular in shape, embedded in abundant silty-sandy matrix. In the thalweg of the Stizzone Creek that runs along the valley, overconsolidated, matrix-supported
diamicton crops out. Different sedimentary, volcanic and metamorphic lithologies are present in the deposit, all showing no evident weathering. These deposits can be ascribed to lodgment and melt-out tills belonging to the Piave glacier. In the nearby Roncon section (Fig. 2), a similar deposit could be observed in few, decametric outcrops at an elevation of about 450 m a.s.l.

### 4.1.2 "Monache" ridges

In the Monache plateau a series of ridges, more than 0.5 km long, is present, at an elevation ranging from 750 to 650 m a.s.l.
(Fig. 2). These ridges are elongated, present a very steep downhill flank, 3 to 20 m high, whilst the uphill side is shorter and locally consists of a low-angle slope. These landforms consist mainly of matrix-supported diamicton characterized by white/grey limestone clasts, up to 50 cm large, with dark flint nodules (black and brown). All observed clasts are made of "Maiolica" limestone, a Lower Cretaceous formation that crops out extensively on the whole Grappa Massif (Dal Piaz et al.,



1946; Carraro et al. 1989), where the ridges are located. These ridges are interpreted as lateral moraines of the Piave glacier fed by local debris of limestones.

### 4.1.3 "Guarda" ridge and core

The Guarda ridge can be found on the left side of the Canal La Menor valley, in the Casere alla Guarda locality, at an elevation ranging from 700 to 660 m a.s.l. (Fig. 2). It is about 400 m long, up to 30 m high and has very steep flanks. This ridge consists of a polygenic diamicton with clasts of porphyry, limestone and siltstone, up to 50 cm in size, sub-rounded to angular in shape. Weathering of clasts is minimal, and the material is not overconsolidated. It is interpreted as a lateral moraine of the tongue of the Brenta glacier that was flowing through the Canal La Menor valley.

A 4.7-m long core (Guarda1 core) was drilled manually in the Guarda locality, at an elevation of 664 m a.s.l., on a wide valley bottom that is closed downstream by the Guarda moraine (Fig. 3 and 6). The cored sequence is constituted by clayey silt layers, brown in color, with sporadic presence of centimetric sandy intervals. A single 30-cm-thick gravelly layer (clasts are 3-4 cm in size) is present between 3.2 and 3.5 cm b.s. The Guarda1 core testifies a phase of low-energy sedimentation in a confined environment, with a single high-energy event that deposited 30 cm of gravel. According to the lack of deep weathering of the deposit and the presence of a poorly-developed soil on top of the succession, the whole sequence is likely to have deposited during the late stages of LGM and/or during the Lateglacial and early Holocene, when the Guarda moraine was blocking the runoff from this lateral valley to the main Canal La Menor valley.

### 4.1.4 "Novegno" and "Col del Gallo" deposits

The top of the Novegno plateau is characterized by a low-gradient plateau at elevations ranging from about 720 to 600 m a.s.l.. In this area, many elongated ridges were found, locally merging one into the other, with a general SW-NE direction (Fig. 2). They are up to 2-km long, 50-m wide and 40-m high, usually with steep slopes. The observed outcrops show that these ridges have polygenic clast composition, being constituted by matrix-supported diamicton with clasts of porphyry, limestone and siltstone, up to 40 cm in size, sub-rounded to angular in shape (Fig. 4b). Between the ridges, as well as on the northern and southern sides of Col del Gallo (780-600 m a.s.l.), scattered patches of similar deposits can be found, characterized by a slightly more abundant content in matrix (Fig. 2, 4c). The deposit is not over-consolidated and the weathering, even of the limestones, is minimal. Locally, some big boulders, up to 1.5-m large, were found; these are mainly of porphyry (Fig. 4a) and granite.

On top of Col del Gallo (870 m a.s.l.), no clear outcrops are present, but centimetric clasts of volcanic and metamorphic phyllites were found at surface.

All evidences show that the Novegno plateau is occupied by many lateral moraines formed by a glacier which was collecting material from at least 25 km north, where the nearest outcrops of porphyry can be found (Fig. 1). Similar glacial sediments are found on the northern and southern sides of Col del Gallo (Fig. 4c). It is likely that they relate to the same glacier. The deposits located on top of Col del Gallo are about 100 m higher than the others and may possibly be related to a previous glacial advance.




### 4.1.5 "Sorist" ridges

In the Sorist mount area a series of at least three ridges, more than 0.8 km long, is present, at an elevation ranging from 760 to 650 m a.s.l. (Fig. 2). These ridges are elongated and present steep flanks, up to 10 m high. Locally the deposits forming these landforms crop out, allowing to determine their polygenic nature, with clasts of porphyry, limestone and siltstone, up to 60 cm

in size, sub-rounded to angular in shape, embedded in abundant matrix (Fig. 4d). Alteration of clasts is minimal and the material is not overconsolidated. Some big porphyry boulders are up to 1 m large.

These ridges are interpreted as left moraines of the LGM Brenta glacier, which was flowing in the underlying Valsugana valley.

### 4.1.6 "Enego" ridge

This 1-km long and more than 70-m high ridge is located on the right side of the Brenta valley, at elevation 790-760 m a.s.l., where the Enego village lies (Fig. 2). This ridge is a lateral moraine of the Brenta glacier, already described by former scholars (Trevisan, 1939; Dal Piaz et al., 1946) and considered to have formed during the LGM. These deposits are the southernmost ones directly connected to the Brenta glacier. During our survey, no large boulders (>1 m) were discovered, but a second, smaller ridge was found at lower elevation (~650 m a.s.l.).

These ridges are interpreted as the right-lateral moraines of the Brenta glacier during the LGM.

### 4.1.7 "Coste" and "Valstagna" sections

In the Coste area, at the bottom of the Valsugana valley on its western side, about 600 m south of the Enego ridge, a stratigraphic section was observed at the excavation front of a gravel pit (Fig. 5), with the base at about 210 m a.s.l. (~10 m above the present valley bottom) (Fig. 2). The section is about 50 m high, 100 m long and it is composed by two main units,

here described:

Lower unit: extending vertically for about 10-15 m, this unit consists of a gravel body rich in sandy matrix, presenting cross-to-planar stratification. Clasts are centimetric in size (maximum diameter is about 30 cm), sub-rounded to sub-angular and consist mainly of carbonate rocks (limestones/marlstones), with abundant siltstones and some granites and porphyries. The lower boundary of this unit is not visible due to the covering by loose debris. The upper boundary with the "upper unit" shows

the interfingering of the two units in the western part of the section, near the foot of the rock wall that constitutes the valley side.

Upper unit: from the top of the lower unit up to the topographic surface there is a sedimentary body made of angular clasts, centimetric-to-decimetric in size, with sandy matrix and scattered boulders (maximum diameter is about 1 m). This unit is crudely stratified, with bedding dipping 25°-30° degrees towards the valley axis. Clasts are lithologically homogeneous,

consisting only of the local carbonate rocks of the overlying rock walls.



The lower unit is ascribable to fluvial deposition by the Brenta River, interfingered and superimposed by scree deposits coming from the overlying rock walls.

An analog sequence has been found on the eastern side of the Brenta valley, in front of the town of Valstagna, about 9 km south of Coste section. Here, a 15-m-high outcrop has been exposed by quarry activity and testify the occurrence of fluvial

deposits (~165 to 170 m a.s.l.), covered by 10 m of slope deposits.

### 4.1.8 "Rocca" deposits

About 1 km northwards of the village of Rocca, two 20-m-high ridges are present, the top being 310 m a.s.l. (Fig. 2). The northernmost one is made of sub-rounded polygenic matrix-supported diamicton with clasts up to 20 cm, sub-rounded to angular in shape. These ridges are interpreted as frontal morainic arcs of a glacier tongue flowing into the Canal La Menor

valley.

The town of Rocca is built upon carbonate bedrock, but a small patch of sediments crops out next to the local graveyard (Fig. 2). This deposit is made of a matrix-supported diamicton with sub-rounded clasts, up to 30 cm large, sub-rounded to angular in shape. Clasts are minimally weathered and mainly carbonate, but porphyry clasts are present as well. The abundant matrix is mainly constituted by sandy particles.

These deposits are interpreted as LGM till belonging to a glacier tongue flowing into the Corlo valley, where the artificial lake is currently located.

### 4.2 Sand petrography

The analysis of the sand fraction in the two cores RB1 and PM1 (Fig. 6) shows three different groups of petrofacies distinguishable both in the main component (Q+F, L, CE) and lithics (Lm, Lv, Ls) ternary diagrams (Fig. 7). Samples related

to petrofacies 1 below the peat layer at 27.5 m b.s. in the RB1 core show high content in quartz, feldspar, felsic volcanic and low-grade metamorphic rock fragments; whereas the carbonate fragments are scarce and around 10 %. Above the peat, the sample RB1-27 shows dolostone clasts to about 25 %, while the felsic volcanic fragments remain high.

Petrofacies 2 have a general content in carbonate clasts generally above 35 % compared to those of petrofacies 1, in particular limestone fragments are common; the other parameters are always abundant, even if with lower percentage.

The single sample of petrofacies 3 shows a completely different composition, with carbonate fragments, especially micritic limestones, up to 55 % and high content in cherts (10 %), normally embedded in the micritic limestones (Barbieri and Grandesso, 2007); on the other hand, this sample has the lower amount of quartz (10 %) and all the other parameters, which are below 10 %.

These results were compared with the present-day sands of the Brenta and Cismon rivers (Garzanti et al., 2006; Monegato et

al., 2010; Fig. 7) through the spreadsheet from Vezzoli and Garzanti (2009). Petrofacies 1 is the most similar to the Brenta River sediments upstream the junction with the Cismon River, except for the sample at 27.3 m b.s. in which dolostones are common (24.3 %): as no main tributaries exist in the lower sector of the Brenta valley, this latter suggests the contribution of





a tributary in the piedmont plain, like the Astico River. Petrofacies 2 is quite similar to the present Brenta River, with an enrichment in carbonate parameters that suggests the contribution of a catchment rich in these components. This input could be from the Piave drainage basin, whereas an input from the Astico-Bacchiglione system in the lowlands can be discarded because during the LGM the river was pushed to the west by the development of the Brenta megafan (Rossato et al., 2013; Fontana et al., 2014), which also managed to dam the Lake Fimon south of Vicenza (Monegato et al., 2011). Finally, petrofacies 3 is remarkable for the high content in carbonate, more similar to the present Piave catchment. However, a direct supply of sediments from such system can be ruled out, as petrofacies 3 is related to a unit sedimented within the post-glacial incision of the Brenta megafan (Mozzi et al., 2013), when the Piave glacier had already collapsed and the Piave River was flowing along its present valley to the East (Pellegrini et al., 2005; Carton et al., 2009). Most of the carbonate clasts are micritic limestone that, coupled with the abundance of cherts, suggest an erosion in the lower Cismon (Corlo) valley north of Rocca (Fig. 1, 3) where these rocks are dominant, or an erosion in the upper Cismon valley close to Lamon (Tessari, 1939).

## 4.3 Mineralogy and geochemistry

Mineralogical analyses of the bulk sediments related to the sedimentary units of the cores (Fig. 6) are reported in terms of main minerals, such as phyllosilicates (mainly micas and chlorites), dolomites (two different crystal chemical terms) and feldspars (plagioclase and k-feldspar). Unit 1 is enriched in feldspars in the diagram reported in Fig. 8, but in its uppermost parts (RB1 and PM1 cores) the distinguishing feature in respect to Unit 2 is the abundance of phyllosilicates, whereas dolomites are depleted. This mineralogical association is very different from that of Unit 3 (very rich in dolomites) and samples of Unit 2 plot in intermediate position are depleted in feldspars. Such features also agree with chemical data (STab. 1), in particular considering $MgO$, $Na_2O$ and $Fe_2O_3$ (SFig. 2), because of their affinity for dolomite, plagioclases and fine-grained minerals (clay minerals, oxides and hydroxides). Calcite is not considered as a discriminating variable because it is strongly depleted or even absent in peat sediments (i.e., > 12 wt % organic matter), making this mineral mainly influenced by the depositional environment and, therefore, poorly indicative of the sediment provenance. Due to this reason, peat samples show higher $Al_2O_3/CaO$ ratio respect to other samples of the same Unit. This chemical ratio for non-peat sediments is a reliable marker of Unit 1, 2 and 3 (SFig. 1). Unit 1 can be also distinguished according to the feldspars/quartz ratio in RB1 core, which is sandier than PM1 core.

## 4.4 Alluvial plain cores

Cores are here subdivided into unformal units based on lithofacies assemblages and petrography, described from the bottom up. The depth of the various layers is referred to the top of the core and indicated with the acronym "b.s." (below surface). Detailed logs are presented in Fig. 6.
Data concerning the samples collected for radiocarbon dating are summarized in Table 1; see the specific core description for details on the position of the samples in the stratigraphy.





### 4.4.1 RB1 core

This borehole was drilled in the Brenta megafan near the town of Carturo, about 5 km north of Piazzola sul Brenta (see Fig. 3 for location, Fig. 6 for stratigraphic log), at a topographic elevation of 30 m a.s.l..

Unit 1: spanning from the bottom of the core up to 27 m b.s., this unit is constituted by sand bodies interbedded with silty layers and two thin (10 cm maximum), fine-grained intervals that are characterized by high organic content. The uppermost organic layer (27.5 m b.s.) was radiocarbon dated to 26.6-27.3 ka cal BP. The sand is normally well sorted, being constituted at maximum by grains 0.5 mm wide. Sandy grains are mainly Quartz, Feldspar and lithic fragments (>30 % of which volcanic) (Fig. 7), with high Feldspar/Quartz (0.6-0.7) and $Al_2O_3/CaO$ (1.3-1.9) ratios in the bulk sediments (SFig. 1).

Unit 2: from 27 to 9 m b.s. there is an alternation of silty and clayey layers with varying sand content. Four very fine-grained organic intervals are present, 6 to 18 cm thick, and two of them has been dated 25.7-26.1 (21.8 m b.s.) and 22.6-23.2 (9.3 m b.s.) ka cal BP, respectively. The base of this unit was placed in correspondence of a sand layer that shows evidence of a change in the petrographic signature. More precisely, in respect to the lower one, this unit shows a remarkable enrichment in carbonates and sedimentary lithic fragments (Fig. 7). Mineralogical and geochemical analyses (bulk) show that the Feldspar/Quartz and $Al_2O_3/CaO$ ratios reduce too (0.25-0.43 and 0.3-1.2, respectively).

Unit 3: the topmost 9 m of RB1 core consists of two different sub-intervals. The bottom one, from 9 up to 2.6 m b.s., has an erosional base and is composed of a coarsening-upward sequence from medium sand to coarse gravel, sub-angular to sub-rounded (largest clasts are about 2 cm). A single, 15 cm-thick silty layer is present at 7.85 m b.s.. The upper sub-interval is remarkably finer, being constituted by a fining upward sequence of sandy and silty layers. No clear evidence of the modern soil has been found at the top. This unit has a clear erosive base, with sand superimposed on clayey silt, which also marks a clear shift in the petrographic, mineralogic and chemical signatures (Fig. 7, 8). This uppermost unit is characterized by the highest content in carbonate (>60 % of the total amount) and sedimentary lithic fragments (Fig. 7). Whilst Feldspar/Quartz and $Al_2O_3/CaO$ ratio values (bulk) are similar to those of Unit 2 (~0.4 and 0.1-0.3, respectively), the dolomite content shows a significant increase (Fig. 8).

Unit 1 is interpreted as sandy proximal overbank deposits, intercalated with more distal overbank fines and intra-ridges peat layers. The fine-dominated succession of Unit 2 indicates low-energy floodplain deposition, with the development of some swampy areas, where organic deposition prevailed over minerogenic contribution, as it was common in the south-eastern alpine piedmont during LGM (Miola et al., 2006; Rossato and Mozzi, 2016). Unit 3 is ascribable to sandy-gravelly channel sediments that eroded the older deposits at the end of the LGM forming fluvial incised valleys through the whole Brenta megafan (Mozzi et al., 2013). The upper part of the unit consists of lower-energy channel deposits or proximal overbank sandy and silty fines.

### 4.4.2 PM1 core

This core was drilled near the town of Piazzola sul Brenta, about 1 km to the west, at an elevation of 27 m a.s.l. (see Fig. 3 for location, Fig. 6 for stratigraphic log).



Unit 1: spanning from the bottom of the core up to 24 m b.s., this unit is constituted by fine grained sand with silty layers. The sand is normally well sorted, being constituted at maximum by grains 0.5 mm large. Sandy grains are mainly Quartz and lithic fragments with bulk Feldspar/Quartz ratio from 0.2 to 0.5 and high $Al_2O_3/CaO$ (1.3-1.9) ratios, in respect to Unit 2, excluding peat sediments.

5 Unit 2: from 24 to the top of the core, this unit is composed by an alternation of silty layers, with a variable content in clay and sand. Two layers of medium sand, located at 19 and 16.3 m b.s. are the only exception to this rather monotonous sequence. Three thin (10 cm maximum) layers with a very high organic content are present in the uppermost 15 m of the succession, two of which have been radiocarbon dated to 23.2-23,7 (13.45 m b.s.) and 19.9-20.4 (9 m b.s.) ka cal BP, respectively. The topmost 2 m of the core show evidence of a well-developed soil, constituted by pedogenic horizons C, Bk and Bw, from the bottom 10 up. The entire core is petrographically homogeneous, being mainly constituted by carbonate (35-40 % of total) and sedimentary lithic fragments (Fig. 7, 8). Bulk $Al_2O_3/CaO$ is lower than 0.7, excluding peats which are depleted in carbonates (STab. 1). As in the core RB1, Unit 1 is interpreted as sandy proximal overbank deposits, while Unit 2 can be interpreted as a fine-dominated succession of floodplain deposition, where more proximal sandy sediments alternate with distal silty layers. Locally, peat deposition took place when minerogenic contribution was very low, as it occurred also in Unit 2 of RB1 core. The topmost 15 soil has characteristic calcic horizons (Calcisol after FAO, 1998), that allow its correlation to the "caranto paleosol", that developed on top of the LGM deposits in the whole Venetian area (Mozzi et al., 2003; ARPAV, 2005; Donnici et al., 2011).

## 4.5 Remote sensing

Whilst image analysis was not very profitable, due to the high vegetation coverage, the DEM provided very valuable data. The DEM allowed to trace laterally those landforms already recognized in the field, where few scattered spots were available, as 20 is the case of the Enego moraine, and to map other new landforms, basing on morphological similarity. In particular, many morainic arcs belonging to the Novegno group have been mapped using this approach. In the geomorphological sketch (Fig. 2), landforms recognized directly on the field are mapped with bright colors, whilst those mapped with remote sensing have fainter shades.

## 5 Discussion

25 Data gathered in the mountain area and in the piedmont megafan are here discussed in order to reconstruct the evolution of the LGM Brenta glacier and tributary glacial systems.

## 5.1 Glaciers in the mountain area

While all the major LGM valley glaciers of the south-eastern Alps preserved all or part of their end-moraine systems in the terminal valley tracts and/or in the piedmont plain (Venzo et al., 1977; Monegato et al., 2007; Carton et al., 2009; Rossato et 30 al., 2013), in the Brenta valley there is no evidence of the LGM (nor older) terminal moraines (Castiglioni, 2004). The Brenta





glacier that used to flow through the Valsugana during the LGM was mainly fed by the Adige glacier, the largest on the southern side of the Alps (Bassetti and Borsato, 2005; Monegato et al., 2017) and some local tributary glaciers from the left valley side. The transfluence of Adige glacier into the Valsugana was through the Fersina saddle (550 m a.s.l.) and the Vigolo Vattaro windgap (lowermost altitude: 680 m a.s.l.); more significantly, above the Calisio plateau (ca 1000 m a.s.l.) during the

maximum glacier expansion. Flowing for about 50 km along the Valsugana, the glacier reached the Primolano sector where the Valsugana narrows from about 1 km to 100 m and the Canal La Menor windgap opens eastwards at about 350 m a.s.l. (Fig. 2).

Basing on our data, the sudden narrowing of the Brenta valley caused the bulging of the glacier inducing it to reach higher elevations up to the Canal La Menor windgap and, thus, to split in two lobes (Fig. 2). The eastern lobe flowing across the

windgap formed the Sorist (760 to 650 m a.s.l.), Novegno (720 to 600 m a.s.l.) and Guarda (700 to 660 m a.s.l.) lateral moraines; the western lobe built the Enego and Col del Gallo (both ~780 m a.s.l.) lateral moraines. The abundance of porphyry clasts and boulders in the deposits related to both eastern and western lobes indicates sediment provenance from the upper Valsugana and the Adige valley. A secondary effect of the glacier bulging would have been the formation of transverse crevasses, producing the fall of supraglacial debris into the ice mass, both hindering the transport of sediments to the glacier

front and increasing the hydraulic conductivity of otherwise effectively impermeable glacier ice (Gulley and Benn, 2007). At the end of the gorge the Valsugana valley widens again, likely inducing the formation of splaying/radial crevasses and icefalls in the frontal glacier mass (Nye, 1952; Harper et al., 1998; Colgan et al., 2016).

The large stratigraphic section in the Coste quarry, and the minor Valstagna one, display no evidence of glacial deposits while it indicates the presence of important LGM glaciofluvial aggradation in front of the western glacier's fronts. Henceforth, the

front of the Brenta glacier flowing in the main valley (western lobe) should have been located between the Coste quarry and the southernmost end of the Enego and Col del Gallo moraines. The presence of the high-elevation lateral moraines hanging above the valley (the Enego moraine is about 550 m higher than the valley floor) suggest that the glacier's front probably consisted in an icefall that hampered the formation of a terminal moraine. This tongue of the Brenta glacier, being a debris-free glacier, has been characterized by a more effective ablation due to solar energy compared to glaciers covered by several

centimeters of debris (Lardeux et al., 2015; Wei et al., 2010). The resulting abundant meltwaters fed a well-developed proglacial stream, inducing aggradation along the whole Brenta valley, as testified by the Coste and Valstagna sections. However, the development of a sublacial/englacial drainage system related to debris-fill crevasses hindered the formation of supraglacial lakes and the seasonal variation of glacier's velocity, thus resulting in a higher resistance to glacier's front fluctuation (Basnett et al., 2013; van der Veen, 2007). Such stability would result in the growing of the Enego and Col del

Gallo lateral moraines.

When the Brenta glacier reached the elevation of the Canal La Menor windgap, the eastern path became an effective glacial flux. The higher the glacier, the more effective would have been the glacial flux through the eastern path, overtopping also the western side of the Novegno plateau. As glacial sediment transport is likely to follow the main glacial flux, so the eastern flow probably subtracted increasingly higher portions of the glacially-transported debris to the western one, depleting the



sedimentary flux through the Valsugana gorge. The eastern glacial lobe flowed along the Canal La Menor windgap down to the confluence with the Corlo valley. Here, it merged with the glacier coming from the north-east, related to the contribution of both the Cismon glacier and the westernmost lobe of the Piave glacier, as testified by the Roncon till. This latter deposit is polygenic, with pebbles of dolostones and Triassic volcanic rocks belonging to the Piave catchment. This latter till crops out

extensively also in the nearby Seren valley, into which a lateral tongue of the Piave glacier was flowing. Geomorphic evidence of the westward flow of Piave glacier is provided by the lateral moraines at Monache (750 to 650 m a.s.l.), even though they are made mostly of limestone clasts, thus reflecting a local glacigenic sedimentary input.

The glacier deriving from the merging of the eastern Brenta lobe and the Cismon/Piave glaciers, at the confluence of Canal La Menor and the Corlo valley, left no traces of frontal moraines. Glacial till crops out close to Rocca (Fig. 2) and till patches

were described about 0.5 km southwards (Dal Piaz et al., 1946), suggesting that the front of this glacier was located at the beginning of the narrow gorge now occupied by the southern end of the artificial Lake Corlo. The geomorphological setting is very similar to the Valsugana one, suggesting that also here it may have formed an icefall with deep crevasses at the glaciers' front.

The Valsugana glacier left a remarkably small amount of erratics. During our survey, a total amount of 7 boulders made of

porphyry and granite, up to 1 m large, have been found (Novegno and Sorist areas). Other authors mentioned erratics on top of the Novegno plateau (up to 2 m large and made of crystalline rocks), next to the Enego moraine ("very large" boulders made of carbonate rocks; Secco, 1883; Venzo, 1940) and on the Canal La Menor valley bottom ("extremely large" porphyry boulders; Taramelli, 1882). No erratics have been found, nor mentioned, downstream along the Valsugana valley bottom.

**5.2 The fluvial record of glaciers' changes**

The fluvial sediments cropping out in the Coste and Valstagna quarries are the only remnants of the LGM glaciofluvial aggradation that took place downstream of both eastern and western glaciers' fronts. This glaciofluvial sedimentation led to the infilling of the Valsugana valley bottom up to some tens of meters above the present Brenta river. The elevation of the top of the LGM valley fill at Coste (about 225 m a.s.l.) is consistent with that one of the Valstagna section (about 175 m a.s.l.) as well as with the top LGM surface of the Brenta megafan SE of Bassano del Grappa (about 130 m a.s.l.). This allows the

correlation of these depositional top surfaces and the related sediments, as well as the reconstruction of the longitudinal profile of the LGM Brenta valley bottom (Fig. 9). The LGM fluvial aggradation was followed by the incision of the valley bottom and the piedmont megafan at around 17.5 ka cal BP, as the fluvial system reacted to the downwasting of the glacial system (Mozzi, 2005; Fontana et al., 2014; Rossato and Mozzi, 2016).

The growth/collapse of glacial tongues and modifications in the fluvial networks can be detected from changes in the mountain

catchments and in the alluvial plain. Sedimentary systems developing at the mouth of major valleys are highly valuable databases of sedimentary, climatic and tectonic data (e.g., Mozzi et al., 2005; Carton et al., 2009; Pini et al., 2009; Piovan et al., 2012; Rossato and Mozzi, 2016). The paucity of radiocarbon datable material usually available in glacial deposits can be balanced with the abundant organic samples that can be collected in fluvial sedimentary sequences of this area. In our





investigation, the petrographic, mineralogical and geochemical and analyses of RB1 and PM1 cores in the glaciofluvially-fed Brenta megafan, chronologically framed through radiocarbon dating, integrate the evidence obtained in the mountain area. This allowed to distinguish specific evolutionary phases in the drainage network feeding the Brenta megafan (Fig. 10), described as follows from the oldest one:

-   The first phase is testified by Unit 1, both in RB1 and PM1 cores. In the former, the top of this unit dates back straight after 27 ka cal BP, when glaciers were growing at the onset of LGM (Monegato et al., 2007; 2017; Ivy-Ochs et al., 2008; Preusser et al., 2011) and the mountain drainage systems began to modify. The Brenta megafan sedimentation rates were still comparable to pre-LGM ones (Rossato and Mozzi, 2016). Sediments indicate that this megafan was fed by a river with a drainage system limited to the Valsugana, with a minor contribution from the Astico system as
shown by the petrographic sample RB1-27. The Cismon drainage system was not merging with the Brenta one at the beginning (petrographic sample RB1-30), probably flowing eastwards into the Piave one (Fig. 10a). Afterwards, the Cismon river started to flow into the Corlo valley, contributing to the Brenta megafan aggradation (petrographic sample RB1-29), before 27 ka cal BP. The trigger responsible for this shift can be looked to, alternatively: i) the rapid and remarkable growth of the Stizzone fan, that pushed the Cismon river to the North until the southern path to the
Corlo valley was favored, compared to the previous eastern one; ii) the arrival of the Cismon glacier at the junction between southern and eastern path that may have perturbed the equilibrium in favor of the Corlo valley path. The first hypothesis seems more reasonable, since it is likely that this shift occurred when glaciers were not grown enough to reach the terminal tracts of the valleys, but an early spread of the Cismon glacier cannot be ruled out.

   -   The base of Unit 2 is marked by a change in the drainage system, as testified by mineralogical and petrographic
analyses (Figs. 7-8) in both cores and spans for the whole LGM. Radiocarbon dates in Piazzola sul Brenta cores indicate that this unit started straight after 27 ka cal BP and continued after 20.1 ka cal BP (Fig. 6). The sedimentary top of Unit 2 in PM1 corresponds to the LGM surface of the Brenta megafan, suggesting that sedimentation lasted until 17.5 ka cal BP (Rossato and Mozzi, 2016). In between 27 and 25.9 ka cal BP, the Piave glacier began to contribute to the Brenta alluvial plain (mineralogic sample RB1-13 - 26.5 m b.s. and petrographic sample RB1- 24-
24.45-24.5 m b.s.; Fig. 6, 10b). This event required the Piave glacier to have grown enough to overcome the Seren saddle (~330 m a.s.l.; Fig. 1), about 100 m above the current Piave valley bottom. Such advanced position of the Piave glacier probably survived the ~19.5 ka cal BP glacial retreat, the Piave glacier still being about 500 m thick near Belluno ("Val Piana" stage: 19,386 - 19,772 years cal BP, recalibrated basing on Pellegrini et al., 2005; Fig. 10c). The Piave system contribution is likely to have blown up the sedimentation rates in the Brenta megafan, that
nearly doubled in the 26.7-23.8 ka cal BP period (Rossato and Mozzi, 2016).

   -   The youngest phase is recorded only in Unit 3 of RB1 core. It is ascribable to the infilling of a fluvial entrenchment (Fig. 2) developed during late stages of deglaciation (Mozzi et al., 2013), as occurred elsewhere in the whole Central and Eastern Po Plain (Fontana et al., 2014b). Mineralogy confirms the present drainage system, with the Cismon River flowing into the Brenta one through the Corlo valley when the Piave River was already flowing into the modern



valley. A remarkable enrichment in carbonates (micritic limestones) in respect of modern Brenta sediments, testified by petrography, mineralogy and geochemistry, pinpoints to an anomalous setting. Two possible scenarios may be proposed to explain such signal: i) a remarkable erosion of such lithologies in the mountain area (i.e. especially in the Cismon catchment), or ii) a connection with the dismantle of the Lamon terraces and/or of the obstacle that induced

their aggradation (if the "landslide scenario" is assumed; see the Setting chapter for details). The first hypothesis can relate with the carving of the Corlo valley by Cismon River during the Lateglacial. Such enhanced erosion stages are likely to occur when meltwater amount is high (Herman et al., 2011), as it is during glacier recessional phases.

## 6 Conclusions

The acquired dataset casts new light on the dynamics of the LGM glaciers in the canyon-like, middle tract of a major Alpine
valley, the Valsugana. The knot of the Valsugana glacier has been disentangled for the first time, indicating a singular configuration of the glacier snouts crossing narrow and deep valley reaches.

Our data indicates that at the LGM acme the Brenta glacier split at Primolano. One tongue used to flow southwards along the Valsugana valley through a narrow gorge which prevented an effective glacier flux and caused the glacier's bulging. The bulging forced the right side of the glacial tongue to reach high elevation at Enego, while its front was constituted by an icefall
located upstream of Coste. Meltwaters were flowing in subglacial and englacial streams filling the Valsugana valley bottom. The other tongue collected most of the glacial and debris fluxes, flowing eastwards along the Canal la Menor valley and joining the Cismon/Piave glacier near Rocca. The front of this glacier probably was an icefall as well, with deep crevasses at its back, located in the Corlo valley.

The coupling of data gathered in the mountain area with those collected in the piedmont alluvial plain allows a coherent
reconstruction of the dynamics of the LGM glacier tongues. Prior to the arrival of the glacier fronts in the study area, the Brenta megafan received sediments only from the upper Valsugana catchment. A major alteration of the drainage system occurred soon before 27 ka cal BP, when the Cismon River abandoned the Piave catchment and joined the Brenta River in the Valsugana valley. It was followed soon after by another major change, directly related to the glacier's dynamics: the arrival of the westernmost tongue of the Piave glacier through the Seren saddle. Between ~27 and, at least, ~19.5 ka cal BP, the Brenta,
Cismon and Piave glaciers were merging in the surroundings of the Novegno mount. Their meltwaters were building up the largest alluvial landform of the whole Venetian-Friulian plain at that time: the Brenta megafan. At the end of LGM, the waning of glaciers induced the fluvial incision of the Brenta megafan. The abundance of micritic carbonates in the sedimentary fill of such incisions near Piazzola sul Brenta suggests a concomitant remarkable bedrock erosion in the Corlo valley and/or the reworking of sediments from the upper Cismon catchment at Lamon.
As general conclusive remarks, this study highlights that:





- the narrowing of a main glaciated valley may result in the blockage/slowing of the glacier flux. A larger lateral valley may easily represent an alternative path for the glacier, even if its bottom lies at higher elevation, subtracting large part of the glacial and debris flux from the main valley;

- valley glaciers flowing across narrow gorges may be subject to bulging and likely have icefalls at their front, which
may prevent the formation of terminal moraines;

- in rugged Alpine terrains, glacial catchments may significantly vary over time during a single glaciation. Such changes affect both the glacial and glaciofluvial systems and can be recognized in the alluvial stratigraphic record far downstream from the glacier front. Sand petrography and chemical/mineralogical composition of sediments are good tracers of glacial catchment variations.

**Author contributions**

All authors contributed to interpreting results and improving the text, that has been written mostly by S. Rossato. Each author contributed to different parts, here listed: field survey: S. Rossato, G. Monegato, P. Mozzi; sand petrography: G. Monegato; X-ray diffraction: A. Carraro, F. Tateo; chemical composition: A. Carraro, F. Tateo; core description: S. Rossato, P. Mozzi; remote sensing: S. Rossato.

**Competing interests**

The authors declare that they have no conflict of interest.

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



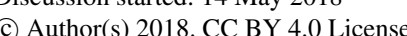

**Figure 1: Geological sketch of the study area. The map is based on the Structural Model of Italy (Bigi et al., 1990) and local geological maps (Bartolomei et al., 1969; Dal Piaz et al., 1946; Barbieri and Grandesso, 2007; Avanzini et al., 2010) and it overlies a SRTM-derived Digital Elevation Model (30-m large cells) [source: http://viewfinderpanoramas.org/].**







**Figure 2: Outcrops map of the middle Valsugana sector derived from field surveys and remote sensing data. Polygons/symbols overlie a 5-m cell DTM (modified from data provided by Regione Veneto, 2011).**



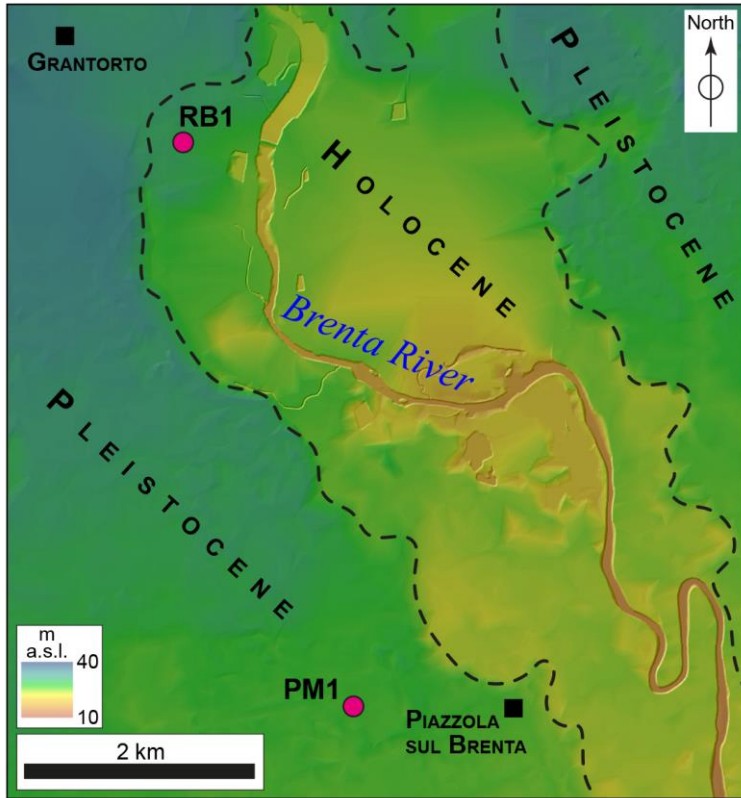

**Figure 3: Location of the PM1 and RB1 cores (purple circles). The background is a 5-m cell DTM (modified from data provided by Regione Veneto, 2011), stretched to highlight elevation changes. Scarps bounding the post-glacial incision of the Brenta megafan are evidenced with black dashed lines.**



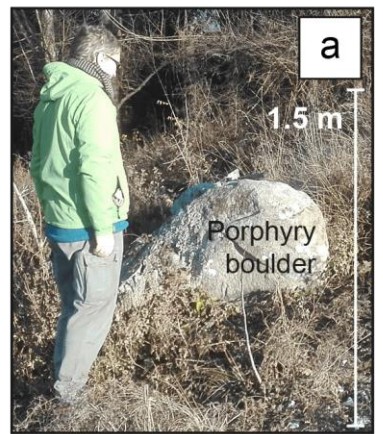

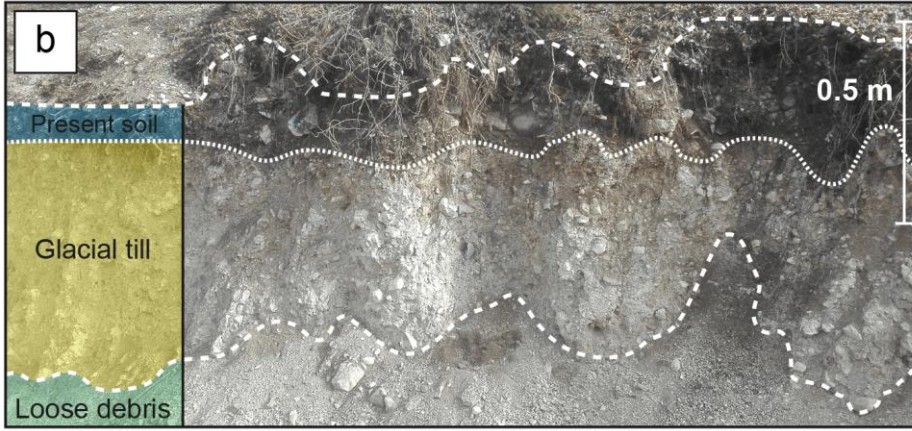

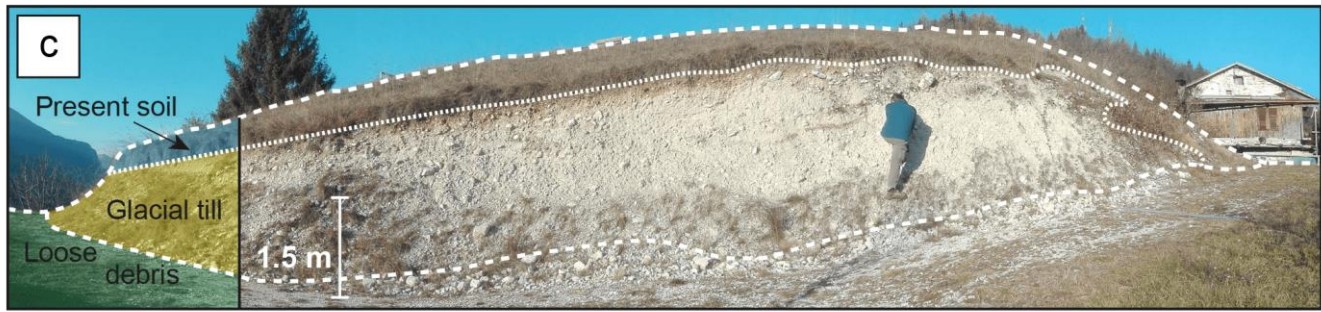

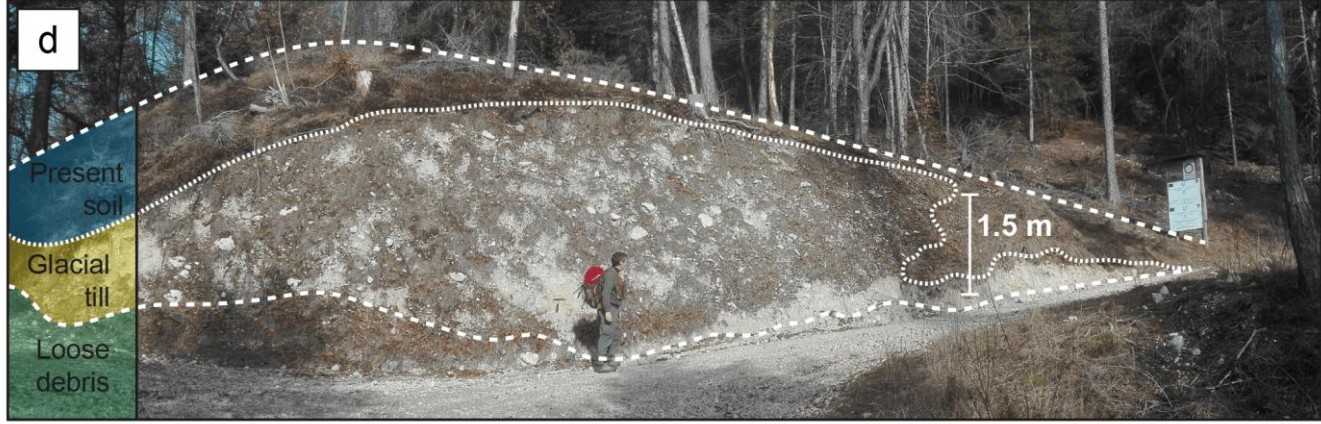

**Figure 4: Photos taken in the middle sector of the Valsugana valley. When present, stratigraphic layers are separated by dotted/dashed white lines. a) porphyry boulder, located on top of Col del Gallo mount; b) section of a lateral moraine of the Brenta glacier, located on top of the Novegno mount; c) moraine, located on the southern sides of Col del Gallo mount; d) moraine, located in the Sorist area.**





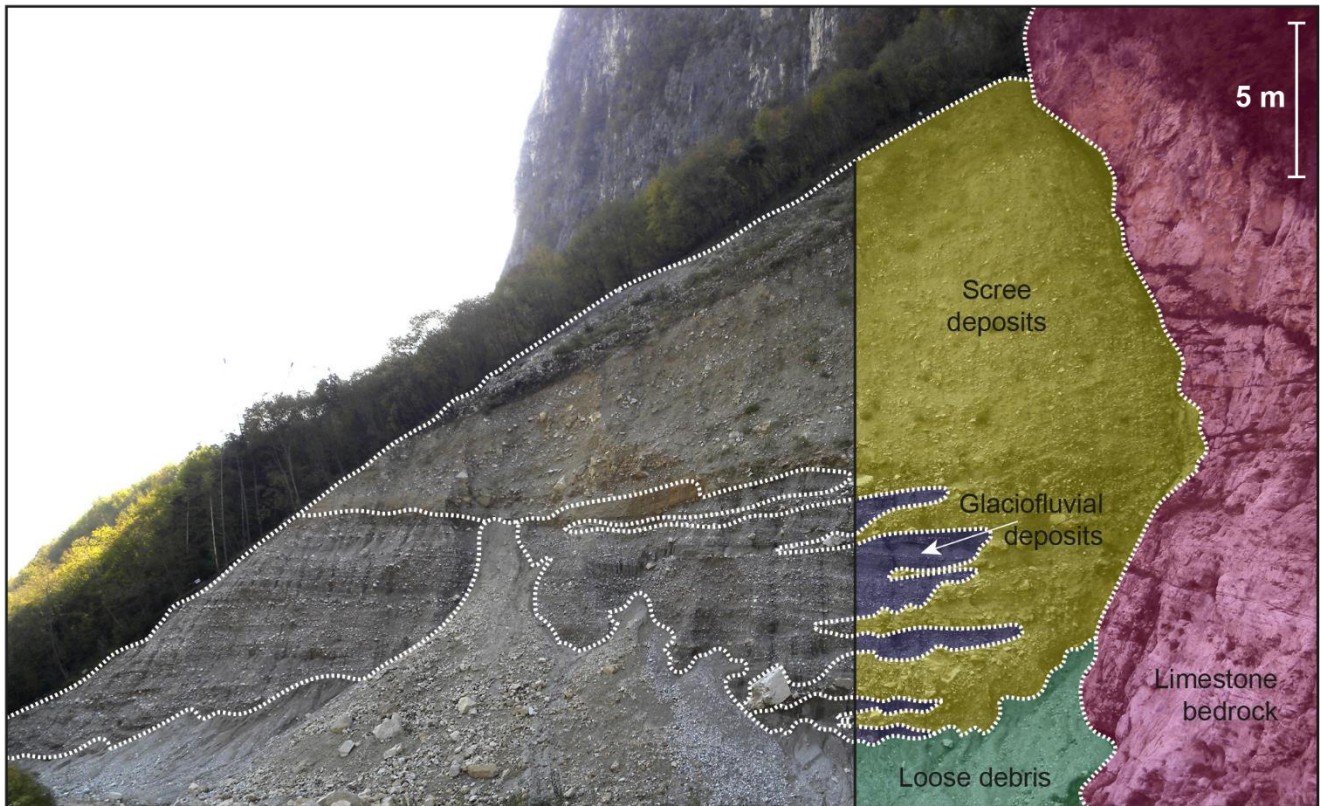

**Figure 5: Coste section. It is possible to appreciate how inclined-bedding scree deposits overlie and interfinger with horizontal-bedding glaciofluvial sediments.**






**Figure 6: Stratigraphic logs of the RB1, PM1 and Guarda1 cores. Stratigraphic units described in the text are evidenced with different colours, while different catchments contributing to the sedimentation are marked by lateral solid/dashed lines. Samples are shown with different symbols according to the adopted technique.**




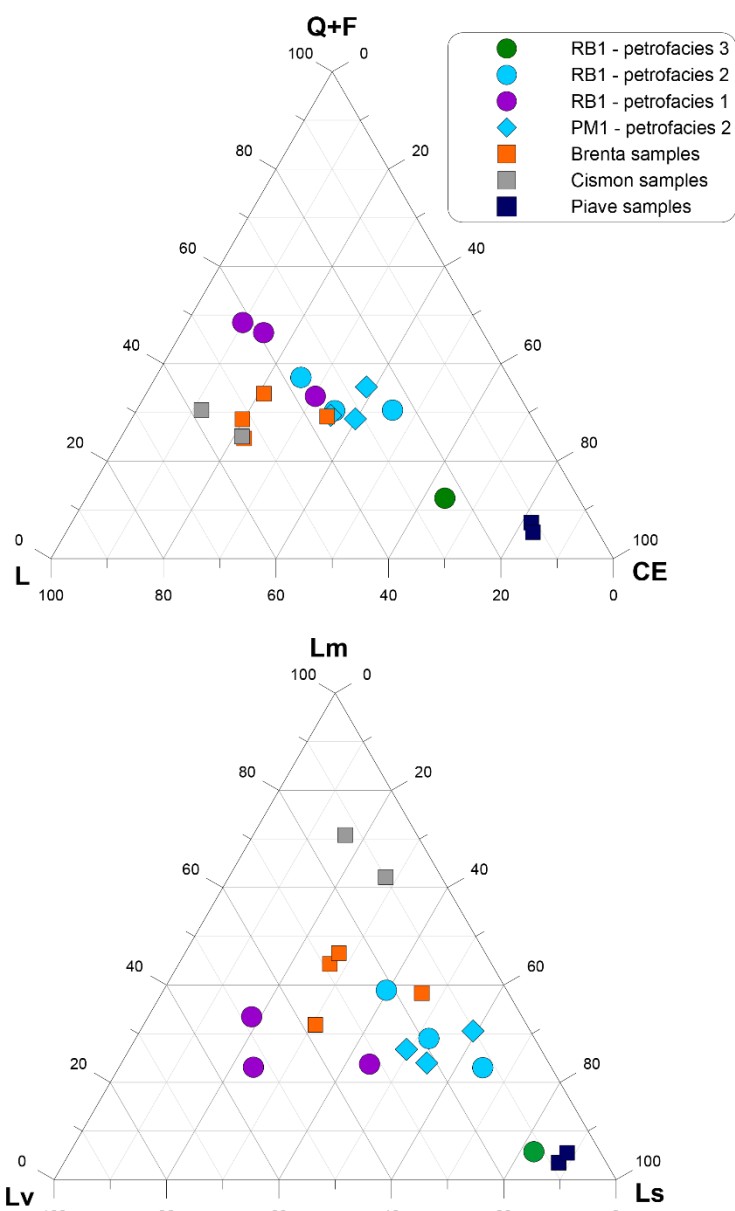

**Figure 7: Ternary diagrams with the results of the sand petrography analysis performed in RB1 and PM1 cores. The compositions of the sediments transported by the present Cismon and Brenta rivers are included.**





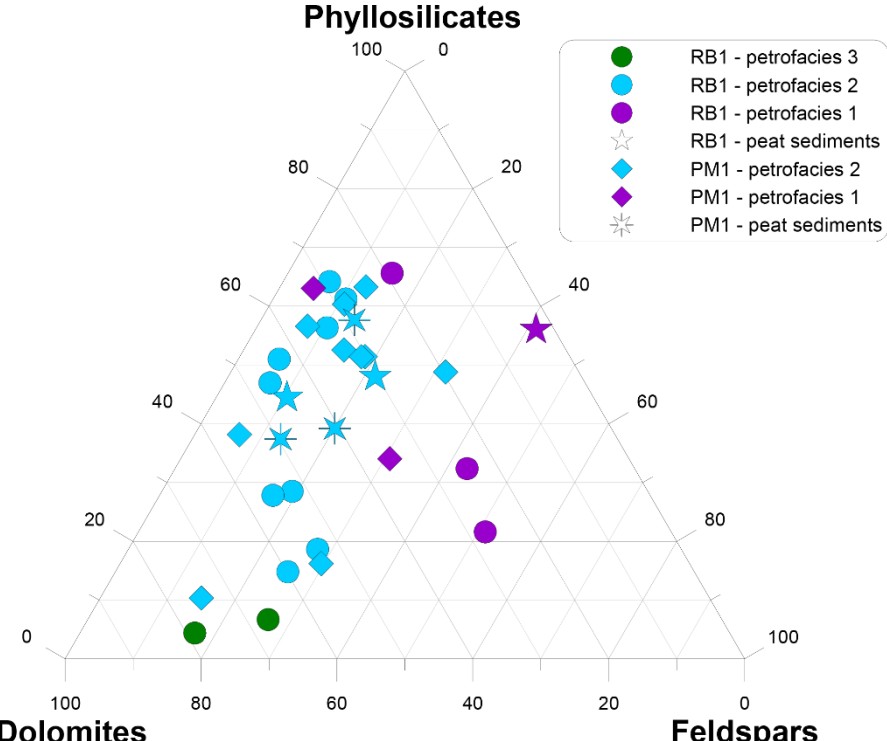

**Figure 8: Ternary diagram reporting main selected mineral components of bulk sediments in RB1 and PM1 cores. Peat samples are evidenced with different symbols.**





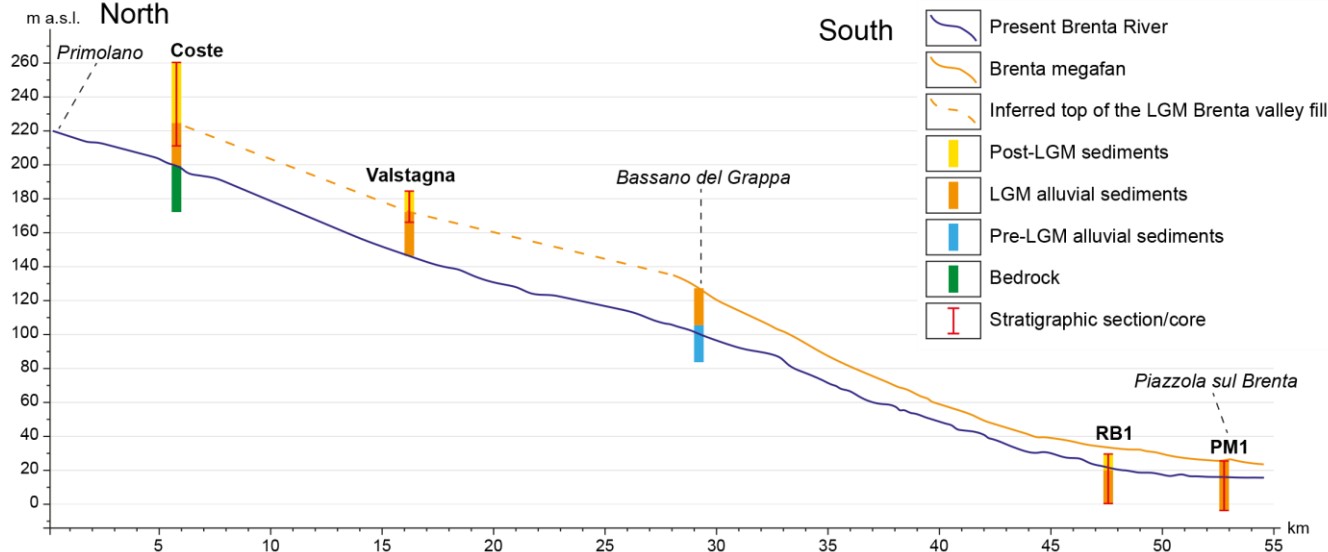

**Figure 9: Longitudinal profiles of the present Brenta river (blue solid line), from Primolano to Piazzola sul Brenta, the Brenta megafan (orange solid line), from the apex to Piazzola sul Brenta, and the possible profile of the Brenta valley bottom during LGM (orange dashed line), inferred from stratigraphic sections. The age of sediments/bedrock is shown with different colours.**



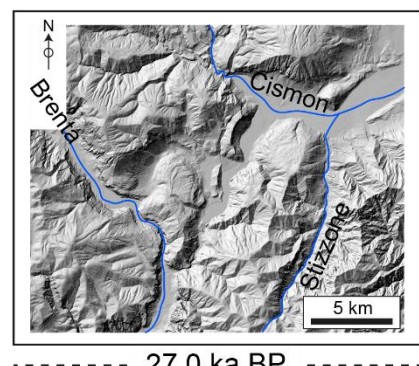

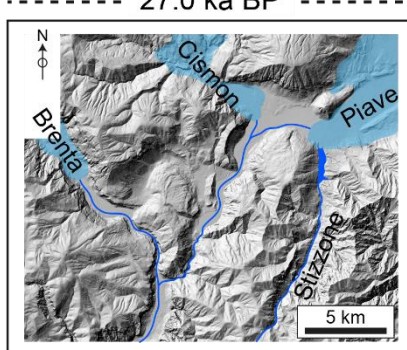

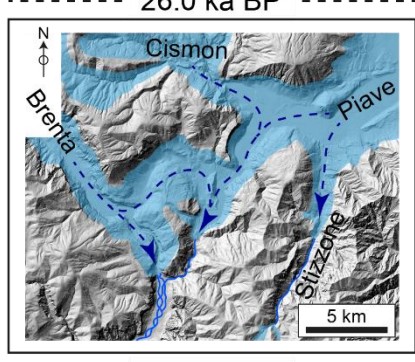

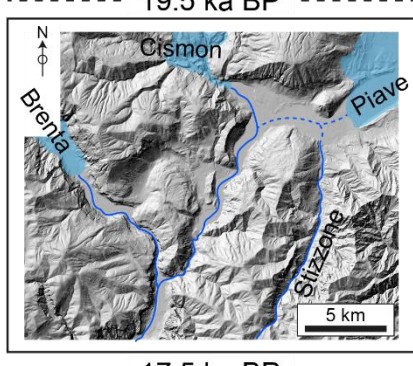

**Figure 10: Evolution of the middle sector of the Valsugana valley since the onset of LGM; see text for details.**





| Sample name | Lab. code | Material | Depth [m] | Uncalibrated age [years BP] | | Calibrated age (IntCal 13 - 2σ) [years BP] | | |
|---|---|---|---|---|---|---|---|---|
| | | | | Lab. age | Uncertainty (±) | Min | Max | Median prob. |
| **PM1-1** | UZ-6073 | Peat | 9.00 | 16,700 | 60 | 19,945 | 20,362 | 20,147 |
| **PM1-2** | UZ-6074 | Peat | 13.45 | 19,500 | 75 | 23,167 | 23,754 | 23,495 |
| **RB1-1** | UZ-6075 | Peat | 9.30 | 19,050 | 70 | 22,648 | 23,221 | 22,938 |
| **RB1-2** | UZ-6076 | Peat | 21.80 | 21,610 | 90 | 25,715 | 26,058 | 25,888 |
| **RB1-3** | UZ-6077 | Peat | 27.50 | 22,660 | 90 | 26,634 | 27,296 | 27,013 |

**Table 1: Conventional, calibrated and median probability [14]C ages obtained from samples collected on RB1 and PM1 cores. Calibration was made with OxCal (version 4.2, Bronk Ramsey, 2009), based on the IntCal13 calibration curve (Reimer et al., 2013).**





| Sample | Depth (m) | Q | F | Lvf | Lvi | Lvp | Lcc | Lcd | Lp | Lch | Lms | Lmi | tot | |
|--------|-----------|-----|------|------|-----|-----|------|------|-----|------|------|-----|-----|---|
| **PM1-12** | 12.1 | 30.5 | 4.7 | 6 | | 0.5 | 14.4 | 20.8 | 3.5 | 0.2 | 19.4 | | 100 | 50%Brenta, 20%Cismon, 30%Piave 0.886 |
| **PM1-17** | 16.6 | 18.2 | 10.6 | 9.1 | 0.3 | 2.6 | 23.9 | 14.8 | 1.4 | 1.1 | 17.9 | 0.3 | 100 | 30%Brenta, 20%Cismon, 50%Piave 0.892 |
| **PM1-19** | 19.4 | 19.5 | 9.9 | 13.1 | | 2.1 | 15.5 | 19.3 | 0.6 | 1.6 | 18.4 | | 100 | 50%Brenta, 20%Cismon, 30%Piave 0.955 |
| **RB1-8** | 7.9 | 10.3 | 2.1 | 8 | | 0.3 | 47.3 | 15.3 | 1.6 | 10.1 | 5.2 | | 100 | Not representative |
| **RB1-16** | 15.7 | 25.1 | 5.5 | 7.5 | | | 18.9 | 22.5 | 4.6 | 0.5 | 15.5 | | 100 | 40%Brenta, 10%Cismon, 50%Piave 0.897 |
| **RB1-21** | 21.4 | 26.8 | 10.4 | 9.7 | | 1.7 | 5.6 | 18.9 | 1.6 | 0.7 | 24.4 | 0.2 | 100 | 60%Brenta, 20%Cismon, 20%Piave 0.953 |
| **RB1-24** | 24.45 | 19.3 | 11.1 | 11.1 | | 1.1 | 12 | 21.6 | 2.2 | 1.8 | 19.7 | 0.2 | 100 | 40%Brenta, 20%Cismon, 40%Piave 0.956 |
| **RB1-27** | 27.3 | 25.1 | 8.3 | 18.8 | 0.3 | | 3.3 | 24.3 | 2.8 | 0.5 | 16 | | 100 | 70%Brenta, 10%Cismon, 20%Astico 0.957 |
| **RB1-29** | 28.95 | 28.3 | 20.2 | 19.2 | | 1.5 | 2.5 | 6.6 | 1.1 | 1 | 19.2 | | 100 | 90%Brenta, 10%Cismon 0.906 |
| **RB1-30** | 29.95 | 27.9 | 18.5 | 21.1 | | 2.6 | 7.8 | 5.3 | 0.3 | 1 | 14.1 | | 100 | 100%Brenta 0.914 |

**Table 2: Detrital modes of the sand fraction collected on RB1 and PM1 cores. List of acronyms: Q: quartz; F: feldspars; Lvf: felsic volcanic and subvolcanic lithic fragments; Lvi: intermediate and mafic lithic fragments; Lvp: plutonic lithic fragments; Lcc: limestone grains; Lcd: dolostone grains; Lp: shale, siltstone lithic fragments; Lch: chert grains; Lms: low-grade metamorphic lithic**
5 **fragments; Lmi: medium-grade metamorphic lithic fragments.**