# Peer review of "Lowlands fluvial sedimentation enlightens glacial dynamics in narrow valleys during the Last Glacial Maximum (Venetian Forealps, Italy)"

_Earth Surface Dynamics, 2018_

## Referee Comment (RC1) · S. Winkler (Referee) · 19 May 2018

General comments:

The discussion article addresses an interesting and important topic. Whereas the focus of research on glacial dynamics and chronology during the LGM in the European Alps naturally has focused on the well-developed moraine sequences major valley glaciers formed when they flowed from their inner Alpine valley sections out onto the foreland, the investigated former Brenta glacier system is of a different type. Confined to a narrow valley it lacks any comparable assemblage of landforms and sediment that would allow easy reconstruction of its outline and chronology, and thus also of its glacial

dynamics. The authors tackle the challenge by applying a multi-proxy approach using different (mainly sedimentological) methods. They, furthermore, aim to link the lithostratigraphical record of the Brenta megafan with the morphological and sedimentological record preserved in the valley. Given the lack of chronological record within the valley itself due to apparent lack of suitable sites and problems of applying surface exposure dating on glacial landforms and bedrock, this seem an appropriate attempt.

The strength of the article is, surely, that a number of different sedimentological methods are combined and that very detailed field work has been carried out to describe and carefully interpret the investigated key sites. The related sedimentological analysis is very sound and altogether the reader can easily follow the argumentation thanks to a number of well-prepared illustrations. The latter is not trivial due to the fact that especially if a study is based on such key sites the reader unfamiliar with those often finds it difficult to assess the detailed interpretation presented in similar stories. This is not the case here.

Finally, the authors develop and discuss some hypothesis about the glacial dynamics of the former Brenta glacier based on their chronological, sedimentological, and chronological findings. Although their conclusions are valid, I have the feeling that alternative explanations could also well be brought forward as reasons to match the evidence presented. In my specific comments below I will address those and want to invite the authors to consider at least mentioning them in the discussion section. Those are hypothesis as well and I, by all means, do not insist that the authors need to change their original interpretation. But by briefly discussing those alternatives (and potentially rejecting them based on their field experience and findings) the authors would show that their explored a wider range of possible explanations. This would further improve the already well presented and written manuscript.

Specific comments:

The authors summarise their conclusions regarding the glacial dynamics within the narrow valley as following. "Glaciers flowing across narrow gorges turned out to be possibly slowed/blocked by such morphology and, if a lateral valley exists, glacial/sediment fluxes can be diverted. Moreover, narrow valleys may induce glaciers to bulge and form icefalls at their front, preventing the formation of terminal moraines".

I have to admit that I am not familiar with the study region and base my comments on my experience in different regions and with modern mountain glaciers. For me, it is not at all surprising that terminal moraines are not present in such relatively narrow mountain valleys. I see, however, not the necessity to infer specific processes like bulging or a specific morphology of the glacier front (like ice falls). Firstly, a plausible explanation for the lack of terminal moraines in the valley is their easy potential erosion. Unlike in the case of LGM valley glacier flowing onto the wide, open forelands a terminal moraine once built is hard to preserve in a setting of a narrow valley where postdepositional glaciofluvial erosion may immediately start eroding the moraine during the initial retreat from the terminal position. Subsequent fluvial erosion (confined to the narrow valley floor) and other geomorphological processes (slope processes etc.) may also contribute to the difficult preservation of terminal moraines and other glacial landforms. By contrast, preservation potential of the major lobate-shaped moraine sequences in the foreland seems much easier as only where Late Glacial or postglacial (glacio)fluvial actions concentrates moraines are easily eroded.

Another explanation for the lack of terminal moraines can be deduced from the different processes of moraine formation. Lateral moraines in high mountain ranges (modern as well as LGM ones) are predominately formed by dumping of supraglacial debris. This well-established mechanism seems undisputed and the less compacted and consolidated character of their glacial diamicts demonstrates this very nicely (as also pointed out by the authors in their description and interpretations of lateral moraine in their study area). With terminal moraine formation there are, however, multiple individual processes involved, partly in complex interaction (ranging from simple pushing to glaciotectonic thrusting). At most modern mountain glacier where terminal moraine has

been studied during the (few) occasions their advanced in more recent decades dumping of debris was either absent or an insignificant contribution to moraine formation. By contrast, most processes that have been identified depend or are at least substantially influenced by the properties of the glacier bed material at the glacier margins, in particular its shear strength and deformability. This applies to situations with unfrozen bed conditions at the glacier margin (pushing) as well as with permafrost at the former glacier margins (glaciotectonic processes). As a result, even with an advancing glacier front no or only a small terminal moraine may be formed if it rest on bedrock or a thin layer of sediment (especially if it has a high shear resistance). Based on these considerations regarding potential moraine formation processes I don't see the necessity to induce any form of "bulging" or a particular morphology of the ice front to explain a lack of terminal moraines. By contrast, I am aware of a modern analogues where a small mountain glacier advanced too fast (but did not surge) to develop a terminal moraine during the ongoing advance where it showed a steep, ice cliff-shaped glacier front. At the time the advances culminated and slowed down, a terminal moraine was pushed up in usual fashion. Summarising, a valid hypothesis for the lack of terminal moraines in the narrow sections of the valley could simply be the different framework (topography, glacier bed material etc.) during culmination of the LGM advance preventing a glacier confined to a narrow valley (possibly with some exposed bedrock at its glacier bed and the lack of deformable soft sediment) to effectively build up a terminal moraine. This hypothesis should be discussed in the related section of the manuscript – and I am more than happy that the authors present evidence that it can be rejected. But currently some readers may ask way the authors did not consider this apparent "easy" and "obvious" solution.

The other explanation offered by the authors that I suggest could be discussed in the light of an alternative explanation is the hypothesis of narrow gorges slowing down/blocking glacier flow and cause diversion. In this context the altitudinal difference between the lateral moraines and the valley floor is additionally mentioned as indication of a blocking action (or bulging) of the glacier flowing through the narrow valley.

Although I can follow the argument given by the authors, it is contradictory to common view that narrow valley channelise ice flow and cause higher flow rates (and increase erosional glacial power). According to some hypothesis promoted by researchers with a background in engineering ice flow mechanics should be seen as comparable with flow mechanics of water. Consequently a certain ice volume transferred from its accumulation area in inner Alpine catchments towards the glacier front as determined by the glacier's mass budget should theoretically speed up if the valley in its flow paths narrows (and not slow down). Any "overspill" and diversion could easily explained by the capacity of the narrow valley not sufficient even with an increased ice flow to transfer the entire ice mass. The huge difference between lateral moraines and valley bottom may indicate that the valley was at the maximum of its capacity with a huge ice mass occupying the valley. In this context, I am also not aware that supraglacial debris (even if potentially integrated into the en- or subglacial debris transport pathways through extensive crevassing) considerably slows down ice flow in those regions currently be the home of extensively debris-covered glaciers. If theoretically a narrow valley inhibits efficient ice flow and obstructs normal mass transfer it would even be a possible cause of glacier surges (that despite multiple theories for their causes all have in common that the normal mass transfer is inhibited until a certain threshold is reached for the surge to start).

I am confident that by exploring the hypothesis mentioned above in the discussion chapter the part of the discussion paper referring to glacial dynamics could be strengthened by providing some alternative views for the author's interpretations of their great field and sedimentological evidence. I am far from insisting that they need to change their conclusions, but feel that currently there is a lack of addressing some common views in the discussion section and some readers may interpret it as some obvious explanations having been overlooked.

Technical corrections:

The manuscript is mostly well structured and written. A few editorial changes may be

**ESurfD**

Interactive
comment

addressed during the revision. I only point out some few points here.

1.) I feel that the title is a bit strong by using the phrase "enlightens glacial dynamics". Even without considering my comments above, there are still some uncertainties that remain. Perhaps the authors could find an alternative title.

2.) In a few sentences, like page 2 line 13 ff., there is an excessive use of commas. Not all are necessary and I would recommend that during the final check of the manuscript, some may be removed.

3. ) The type of radiocarbon-dated material and its position are given in the related table. I only miss information about the sampled thickness (I assume 1 cm?).

4.) Wasn't it possible to asses a potential difference between lodgement and melt-out till and make a judgement here?

---

## Referee Comment (RC2) · L. Stutenbecker (Referee) · 23 May 2018

A) General comments

The discussion paper by Sandro Rossato and co-authors presents an interesting approach to infer LGM dynamics in a narrow valley in the southern Alps, mainly using a provenance tracing technique applied to sediments in the corresponding lowland. Overall the paper is well structured and it is easy to follow the central theme. The methodological description of the provenance tracing approach could be improved with regards to the following 3 points:

[Figure]

1) I feel that the explanation of the mixing modeling approach used to infer the relative contributions of the modern Brenta, Cismon and Piace rivers to the sediments is a bit short. Sure, your approach basically uses the same strategy as described in Vezzoli & Garzanti (2009) and the river endmembers defined in Garzanti et al. (2006), but it needs to be explained a bit more in detail (goodness of fit, errors, etc.).

2) In sections 4.2 to 4.4 I couldn't follow the definition of the "petrofacies" and the "units", respectively. Were the "units" of the cores defined based on the petrofacies? Or the other way round? The text is a bit ambiguous in this regard and needs clarifying. Perhaps it would make sense to describe the cores first (section 4.4) and then interpret the petrography/mineralogy/geochemistry (sections 4.2 and 4.3)?

3) I would highly recommend using principal component analysis (PCA) for the interpretation and visualization of the petrographical, mineralogical and chemical datasets. PCA has become a standard tool in provenance analysis and the compositional biplots really help visualizing differences between samples and identifying clusters (see e.g. Aitchison, 1982, Biometrika; Aitchison & Greenacre, 2002, Applied Statistics; Vermeesch, 2013, Chemical Geology). The ternary plots are okay to use for a first visualization, but I wonder which additional conclusions could be drawn from a compositional biplot. You could for example try plotting all parameters together (petrography, mineralogy, chemistry) to see how the clustering goes. Don't forget to also plot your modern river endmembers.

I would recommend the CoDaPack from the group at the University of Girona: http://www.compositionaldata.com/codapack.php (reference to use: Comas, M., Thió-Henestrosa, S., 2011. CoDaPack 2.0: a stand-alone multi-platform compositional software. In: Egozcue, J.J., Tolosana-Delgado, R., Ortego, M.I. (Eds.), CoDaWork'11: 4th International Workshop on Compositional Data Analysis. Saint Feliu de Guixols, Girona, Spain). You just import your table as a .csv or .txt file and then go to Graphs/centered-log-ratio (CLR) biplot. Alternatively, if you like playing with R, you might consider using the "Provenance" toolbox of Pieter Vermeesch (Vermeesch, P.,

Resentini, A. and Garzanti, E., 2016. An R package for statistical provenance analysis. Sedimentary Geology, 336, 14-25)

B) Detailed comments (page and line numbers refer to the online version of the manuscript)

p. 2 line 20: "allow FOR" instead of "allow to TO"

p. 2 line 32: there is no plural for "evidence".

p. 3 line 17: "Last glaciation". Either you use Last Glaciation (both capital letters) if you use this as a proper noun or "last glaciation" without any capital letters.

p. 3 line 20: Add "(Fig. 1)" after "left well-preserved terminal moraines".

p. 4 line 12: use either the singular or the plural for the rock types. I suggest to use "porphyries" instead of "porphyry" here

p. 4 lines 11-14: This paragraph about the geology doesn't really fit into "3.1 Field survey". Either you move this paragraph to "2 Setting" if it's just a description of the geology or into another heading if your aim is to describe your provenance strategy.

I am a bit confused by the "granites, porphyries and metamorphic rocks" you mention. In Fig. 1 you only show group of rocks ("volcanic" or "plutonic"), which is understandable in order for the figure to be legible, but into which groups do the "granites, porphyries and metamorphic rocks" belong? I guess the metamorphic rocks belong to the Variscan basement and the porphyries into the "Permian volcanic rocks"?

I guess you mean that the Brenta drainage area comprises plutonic and volcanic rocks, which are not present in the neighboring Astico and Piave valleys, and that this difference makes it possible to identify the respective deposits? Please make all of this clearer by writing 2-3 more sentences.

p. 5 line 2: "0.0625-2" Did you use this particular grain size fraction in order to compare your dataset with the river endmembers from Garzanti et al. (2006)? If so, please state

this in the methods. By taking such a wide grain size window one risks to introduce bias by grain size sorting. . .

p. 5 lines 4-5: Please rephrase the sentence about the point counting, for example "Following the Gazzi-Dickinson method 400 points per thin section were counted using a 0.5 mm grid spacing (Ingersoll et al. 1984)." Did you use the same grain classes as Vezzoli & Garzanti (2009)? If so, please state so in this paragraph.

p. 5 line 6 "Data and parameters were reported in Table 2 AND plotted in ternary diagrams."

p. 5 lines 6-9: Please describe more thoroughly the strategy behind this approach (defining endmembers, applying a linear mixing model, reporting the goodness of fit,. . ..)

p. 5 line 12: Avoid the word "adopt" in this context.

p. 7 line 4: left side? Please use geographic directions (west/east)

p. 7 line 27: "found at THE surface"

p. 7 line 28: "Evidence" has no plural: "All evidence shows. . ."

p. 7 line 29: ". . .which was collecting material from an area located at least 25 km to the north. . ."

p. 8 line 10: right side? Please use geographic directions (west/east)

p. 9 line 1: "The lower unit could be attributed to. . ."

p. 9 line 19: What is "CE"? I couldn't find it in the text.

p. 9 line 22: ". . .while the content of felsic volcanic fragments remains high."

p. 9 line 23-24: Please rephrase to something like "Although the spectrum of lithic fragments contained in petrofacies 2 is similar to that of petrofacies 1, petrofacies 2 contains more carbonate clasts, generally above 35%. Micritic limestone fragments

are particularly common."

p. 9 lines 25-28. This is a long and convoluted sentence. Rephrase to something like "The single sample of petrofacies 3 shows a completely different composition. The carbonate fragment content increases to 55% at the expense of quartz (only 10 %) and other grain types (below 10 %).". . . Also I do not understand how the chert is embedded into the limestone. . . is it a partially dissolved and then recrystallized limestone or . . .? Please specify by including a better petrographic description.

p. 10 line 2: ". . .with an enrichment of carbonate rock fragments. . ."

p. 10 lines 5-6: "Finally, petrofacies 3 with its high carbonate clast content is more similar to the modern Piave River sediment."

p. 10 lines 6-9: Couldn't this be interpreted simply as a reworking of deposits from the Piave catchment?

p. 10 lines 13-25: Do "unit 1" and "unit 2" refer to "petrofacies 1" and "petrofacies 2" from before? If so, please use the same name, either "petrofacies" or "unit".

p. 11 lines 7-8: No capital letters for quartz and feldspar!

p. 11 line 10: . . ." and two of them have been dated"

p. 11 line 14: Again, no capital letters for quartz and feldspar.

p. 11 line 21: Again, no capital letters for quartz and feldspar.

p. 11 lines 22-23: ". . . the dolomite content is significantly higher (Fig. 8)."

p. 12 lines 2-3: Again, no capital letters for quartz and feldspar.

p. 12 line 28: parts

p. 13 line 2: left valley side? Please use geographic directions (west/east)

p. 13 line 8: "Based" instead of "Basing"

p. 13 lines 8-11: I do not understand this interpretation... Which of your data supports this? Please specify

p. 13 line 23: consisted of

p. 13 line 33: the "so" in the sentence can be deleted

p. 15 line 13: Not sure what you mean by "looked to"? Linked to?

p. 16 line 1: in respect to

C) References

I did not thoroughly check all the references, but there are at least two where author's names are not capitalized (e.g. page 19 line 23 "Andò" or page 20, line 1 "Anderson").

D) Comments on figures and tables

Figure 1: In the legend you use UPPER Permian for the sandstones but EARLY Permian for the plutonic rocks. Use either "Upper and Lower" or "Early and Late" to make this consistent. See for instance Haile 1987 (Marine and Petroleum Geology) for the use of this nomenclature.

Figure 2: Increase the size of the yellow square indicating the drill site.

Figure 6: What is the red square in the uppermost left corner and why is it red?

Figure 7: Add to the figure caption the explanations of the ternary corners (CE, Lm, Lv, Ls...). Did you group together certain grain classes?

Table 2: Add a heading for the last column of this table (e.g. "Relative contribution of endmembers" or something like that). Do the numbers (0.886 and so on) refer to R2? Please explain this! What's up with sample RB1-8? Why is it "not representative"?

---

## Author Comment (AC1) · 24 Jul 2018

We would like to thank both reviewers for their comments. We surely think that the manuscript has been considerably improved by such pieces of advice. Here below we reply to reviewers. We grouped comments by reviewer #1 at the beginning (we split the comment according to the topic and marked each section as "RC1"), followed by reviewer #2 ones ("RC2"). Our replies are indicated with the acronym "AC". As a supplement to this comment you can find a zip archive with two PDFs: the revised manuscript and the revised manuscript with highlighted the changes in respect to the pre-revision version.

[Figure]

REFEREE #1: S. Winkler

A) General comments

RC1: The discussion article addresses an interesting and important topic. Whereas the focus of research on glacial dynamics and chronology during the LGM in the European Alps naturally has focused on the well-developed moraine sequences major valley glaciers formed when they flowed from their inner Alpine valley sections out onto the foreland, the investigated former Brenta glacier system is of a different type. Confined to a narrow valley it lacks any comparable assemblage of landforms and sediment that would allow easy reconstruction of its outline and chronology, and thus also of its glacial dynamics. The authors tackle the challenge by applying a multi-proxy approach using different (mainly sedimentological) methods. They, furthermore, aim to link the lithostratigraphical record of the Brenta megafan with the morphological and sedimentological record preserved in the valley. Given the lack of chronological record within the valley itself due to apparent lack of suitable sites and problems of applying surface exposure dating on glacial landforms and bedrock, this seem an appropriate attempt. The strength of the article is, surely, that a number of different sedimentological methods are combined and that very detailed field work has been carried out to describe and carefully interpret the investigated key sites. The related sedimentological analysis is very sound and altogether the reader can easily follow the argumentation thanks to a number of well-prepared illustrations. The latter is not trivial due to the fact that especially if a study is based on such key sites the reader unfamiliar with those often finds it difficult to assess the detailed interpretation presented in similar stories. This is not the case here. Finally, the authors develop and discuss some hypothesis about the glacial dynamics of the former Brenta glacier based on their chronological, sedimentological, and chronological findings. Although their conclusions are valid, I have the feeling that alternative explanations could also well be brought forward as reasons to match the evidence presented. In my specific comments below I will address those and want to invite the authors to consider at least mentioning them in the discussion section. Those

are hypothesis as well and I, by all means, do not insist that the authors need to change their original interpretation. But by briefly discussing those alternatives (and potentially rejecting them based on their field experience and findings) the authors would show that their explored a wider range of possible explanations. This would further improve the already well presented and written manuscript.

AC: We are glad that the reviewer appreciated our efforts and thank him for the useful comments. Here below we reply to them:

B) Specific comments:

RC1: The authors summarise their conclusions regarding the glacial dynamics within the narrow valley as following. "Glaciers flowing across narrow gorges turned out to be possibly slowed/blocked by such morphology and, if a lateral valley exists, glacial/sediment fluxes can be diverted. Moreover, narrow valleys may induce glaciers to bulge and form icefalls at their front, preventing the formation of terminal moraines". I have to admit that I am not familiar with the study region and base my comments on my experience in different regions and with modern mountain glaciers. For me, it is not at all surprising that terminal moraines are not present in such relatively narrow mountain valleys. I see, however, not the necessity to infer specific processes like bulging or a specific morphology of the glacier front (like ice falls). Firstly, a plausible explanation for the lack of terminal moraines in the valley is their easy potential erosion. Unlike in the case of LGM valley glacier flowing onto the wide, open forelands a terminal moraine once built is hard to preserve in a setting of a narrow valley where postdepositional glaciofluvial erosion may immediately start eroding the moraine during the initial re-treat from the terminal position. Subsequent fluvial erosion (confined to the narrow valley floor) and other geomorphological processes (slope processes etc.) may also contribute to the difficult preservation of terminal moraines and other glacial landforms. By contrast, preservation potential of the major lobate-shaped moraine sequences in the foreland seems much easier as only where Late Glacial or postglacial (glacio)fluvial actions concentrates moraines are easily eroded. Another explanation for the lack of

terminal moraines can be deduced from the different processes of moraine formation. Lateral moraines in high mountain ranges (modern as well as LGM ones) are predominately formed by dumping of supraglacial debris. This well-established mechanism seems undisputed and the less compacted and consolidated character of their glacial diamicts demonstrates this very nicely (as also pointed out by the authors in their description and interpretations of lateral moraine in their study area). With terminal moraine formation there are, however, multiple individual processes involved, partly in complex interaction (ranging from simple pushing to glaciotectonic thrusting). At most modern mountain glacier where terminal moraine has been studied during the (few) occasions their advanced in more recent decades dumping of debris was either absent or an insignificant contribution to moraine formation. By contrast, most processes that have been identified depend or are at least substantially influenced by the properties of the glacier bed material at the glacier margins, in particular its shear strength and deformability. This applies to situations with unfrozen bed conditions at the glacier margin (pushing) as well as with permafrost at the former glacier margins (glaciotectonic processes). As a result, even with an advancing glacier front no or only a small terminal moraine may be formed if it rest on bedrock or a thin layer of sediment (especially if it has a high shear resistance). Based on these considerations regarding potential moraine formation processes I don't see the necessity to induce any form of "bulging" or a particular morphology of the ice front to explain a lack of terminal moraines. By contrast, I am aware of a modern analogues where a small mountain glacier advanced too fast (but did not surge) to develop a terminal moraine during the ongoing advance where it showed a steep, ice cliff-shaped glacier front. At the time the advances culminated and slowed down, a terminal moraine was pushed up in usual fashion. Summarising, a valid hypothesis for the lack of terminal moraines in the narrow sections of the valley could simply be the different framework (topography, glacier bed material etc.) during culmination of the LGM advance preventing a glacier confined to a narrow valley (possibly with some exposed bedrock at its glacier bed and the lack of deformable soft sediment) to effectively build up a terminal moraine. This hypothesis

**ESurfD**

Interactive
comment

should be discussed in the related section of the manuscript – and I am more than happy that the authors present evidence that it can be rejected. But currently some readers may ask way the authors did not consider this apparent "easy" and "obvious" solution.

AC: We thank the reviewer for the detailed comment and prompts for discussion. We added in the text the proposed scenarios, discussing them in the light of our data. All in all, we think that the most probable explanation is that material brought to the front, both during the advance and stability phases, was completely carried away by proglacial streams. The location of the glacier front is constrained by lateral moraines and the Coste fluvioglacial deposits within about 250 m. The vertical drop in elevation in such a small distance between moraines and present valley floor (about 550 m) fits well with an icefall front. Being laterally constrained by hard rock steep valley flanks, proglacial fluvial processes could remove debris progressively, hindering the formation of end moraines. Bulging of the western lobe in respect of the eastern one is suggested by the location of the various morainic arcs that are about 20-to-100 m higher in elevation to the west.

RC1: The other explanation offered by the authors that I suggest could be discussed in the light of an alternative explanation is the hypothesis of narrow gorges slowing down/blocking glacier flow and cause diversion. In this context the altitudinal difference between the lateral moraines and the valley floor is additionally mentioned as indication of a blocking action (or bulging) of the glacier flowing through the narrow valley. Although I can follow the argument given by the authors, it is contradictory to common view that narrow valley channelise ice flow and cause higher flow rates (and increase erosional glacial power). According to some hypothesis promoted by researchers with a background in engineering ice flow mechanics should be seen as comparable with flow mechanics of water. Consequently a certain ice volume transferred from its accumulation area in inner Alpine catchments towards the glacier front as determined by the glacier's mass budget should theoretically speed up if the valley in its flow paths nar-

rows (and not slow down). Any "overspill" and diversion could easily explained by the capacity of the narrow valley not sufficient even with an increased ice flow to transfer the entire ice mass. The huge difference between lateral moraines and valley bottom may indicate that the valley was at the maximum of its capacity with a huge ice mass occupying the valley. In this context, I am also not aware that supraglacial debris (even if potentially integrated into the en- or subglacial debris transport pathways through extensive crevassing) considerably slows down ice flow in those regions currently be the home of extensively debris-covered glaciers. If theoretically a narrow valley inhibits efficient ice flow and obstructs normal mass transfer it would even be a possible cause of glacier surges (that despite multiple theories for their causes all have in common that the normal mass transfer is inhibited until a certain threshold is reached for the surge to start). I am confident that by exploring the hypothesis mentioned above in the discussion chapter the part of the discussion paper referring to glacial dynamics could be strengthened by providing some alternative views for the author's interpretations of their great field and sedimentological evidence. I am far from insisting that they need to change their conclusions, but feel that currently there is a lack of addressing some common views in the discussion section and some readers may interpret it as some obvious explanations having been overlooked.

AC: We added some sentences exploring the aspects highlighted by the reviewer. As mentioned by him, valley narrowing is known to speed up glacier flowing due to ice mechanics. Nonetheless, in some cases, remarkable reductions of the valley section are known to have caused the blockage of glaciers (Burbank and Fort, 1985). In our case, the valley section reduces of about 90% (from about 1 km to 100 m), thus we consider that friction at the glacier margin probably slowed down/blocked the ice flow. This situation must have lasted only for a limited amount of time: as soon as Canal La Menor became an effective path for the glacier, the western tongue became unprivileged and is likely to have stopped almost completely. Indeed, glacial/sediment fluxes can be diverted when stabilized valley glaciers can extend laterally (Barr and Lovell, 2014). Glacial surges occurred prior of the activation of the Canal La Menor, if ever

occurred, left no traces in the sedimentary record and in the valley morphology. Moreover, the glacial front after such surge must have withdrawn to the Valsugana gorge in order to let the Coste section to form. That said, it seems unlikely to us that such surges may have occurred.

C) Technical corrections:

RC1: The manuscript is mostly well structured and written. A few editorial changes may be addressed during the revision. I only point out some few points here. 1.) I feel that the title is a bit strong by using the phrase "enlightens glacial dynamics". Even without considering my comments above, there are still some uncertainties that remain. Perhaps the authors could find an alternative title.

AC: We tried to find a new title to both "soften" the previous one and maintain the focus. Here is our attempt: "Glacial dynamics in Prealpine narrow valleys during the Last Glacial Maximum inferred by lowlands fluvial record (NE Italy)".

RC1: 2.) In a few sentences, like page 2 line 13 ff., there is an excessive use of commas. Not all are necessary and I would recommend that during the final check of the manuscript, some may be removed.

AC: Also referee #2 found pointed out some grammar and style suggestions. We re-read completely the text, trying to improve it.

RC1: 3. ) The type of radiocarbon-dated material and its position are given in the related table. I only miss information about the sampled thickness (I assume 1 cm?).

AC: A column has been added to that table, containing the requested information.

RC1: 4.) Wasn't it possible to asses a potential difference between lodgement and melt-out till and make a judgement here?

AC: We modified the description to better clarify the differences between lodgment and melt-out till. In our case, the differences between them are the clast shape (more

**ESurfD**
rounded in the lodgment till) and the matrix (silty in the lodgment till and sandy-silty in the melt-out till).

REFEREE #2: L. Stutenbecker

A) General comments

RC2: The discussion paper by Sandro Rossato and co-authors presents an interesting approach to infer LGM dynamics in a narrow valley in the southern Alps, mainly using a provenance tracing technique applied to sediments in the corresponding lowland. Overall the paper is well structured and it is easy to follow the central theme. The methodological description of the provenance tracing approach could be improved with regards to the following 3 points:

AC: We thank the reviewer for the useful comments, we found them very useful. Here we reply to them, one by one:

RC2: 1) I feel that the explanation of the mixing modeling approach used to infer the relative contributions of the modern Brenta, Cismon and Piace rivers to the sediments is a bit short. Sure, your approach basically uses the same strategy as described in Vezzoli & Garzanti (2009) and the river endmembers defined in Garzanti et al. (2006), but it needs to be explained a bit more in detail (goodness of fit, errors, etc.).

AC: We improved both in the methods and in the description the strategy of the mixing provenance after Vezzoli and Garzanti (2009).

RC2: 2) In sections 4.2 to 4.4 I couldn't follow the definition of the "petrofacies" and the "units", respectively. Were the "units" of the cores defined based on the petrofacies? Or the other way round? The text is a bit ambiguous in this regard and needs clarifying. Perhaps it would make sense to describe the cores first (section 4.4) and then interpret the petrography/mineralogy/geochemistry (sections 4.2 and 4.3)?

AC: We decided to avoid the use of "unit" and re-arranged the chapter moving core description to the beginning, followed by petrography/mineralogy/geochemistry results

and interpretations. "Petrofacies" remains the only subdivision criterium without strati-graphic ambiguity.

RC2: 3) I would highly recommend using principal component analysis (PCA) for the interpretation and visualization of the petrographical, mineralogical and chemical datasets. PCA has become a standard tool in provenance analysis and the compositional biplots really help visualizing differences between samples and identifying clusters (see e.g. Aitchison, 1982, Biometrika; Aitchison & Greenacre, 2002, Applied Statistics; Vermeesch, 2013, Chemical Geology). The ternary plots are okay to use for a first visualization, but I wonder which additional conclusions could be drawn from a compositional biplot. You could for example try plotting all parameters together (petrography, mineralogy, chemistry) to see how the clustering goes. Don't forget to also plot your modern river endmembers. I would recommend the CoDaPack from the group at the University of Girona: http://www.compositionaldata.com/codapack.php (reference to use: Comas, M., Thió- Henestrosa, S., 2011. CoDaPack 2.0: a stand-alone multi-platform compositional software. In: Egozcue, J.J., Tolosana-Delgado, R., Ortego, M.I. (Eds.), CoDaWork'11: 4th International Workshop on Compositional Data Analysis. Saint Feliu de Guixols, Girona, Spain). You just import your table as a .csv or .txt file and then go to Graphs/centered-log-ratio (CLR) biplot. Alternatively, if you like playing with R, you might consider using the "Provenance" toolbox of Pieter Vermeesch (Vermeesch, P., Resentini, A. and Garzanti, E., 2016. An R package for statistical provenance analysis. Sedimentary Geology, 336, 14-25)

AC: We used the PCA (CoDaPack 2.0) for the interpretation of the petrographical dataset because we have endmembers to plot. We presented the results with the new diagrams in Fig. 7. We tried to use PCA also for visualization and interpretation of mineralogical and geochemical data, but clusters identified for different petrofacies have not substantially changed. Moreover, lacking appropriate data concerning mineralogical and geochemical analyses of end members (i.e., different lithologies of modern river endmembers), a complete provenance evaluation cannot be done. For these reasons, we prefer to maintain conventional descriptive diagrams, which are useful for a first visualization but also evidence the main distinctive features of petrofacies.

B) Detailed comments

AC: minor comments are grouped together where no specific clarification is needed. More detailed explanations are given where necessary.

RC2: p. 2 line 20: "allow FOR" instead of "allow to TO" p. 2 line 32: there is no plural for "evidence". p. 3 line 17: "Last glaciation". Either you use Last Glaciation (both capital letters) if you use this as a proper noun or "last glaciation" without any capital letters. p. 3 line 20: Add "(Fig. 1)" after "left well-preserved terminal moraines". p. 4 line 12: use either the singular or the plural for the rock types. I suggest to use "porphyries" instead of "porphyry" here

AC: Done.

RC2: p. 4 lines 11-14: This paragraph about the geology doesn't really fit into "3.1 Field survey". Either you move this paragraph to "2 Setting" if it's just a description of the geology or into another heading if your aim is to describe your provenance strategy. I am a bit confused by the "granites, porphyries and metamorphic rocks" you mention. In Fig. 1 you only show group of rocks ("volcanic" or "plutonic"), which is understandable in order for the figure to be legible, but into which groups do the "granites, porphyries and metamorphic rocks" belong? I guess the metamorphic rocks belong to the Variscan basement and the porphyries into the "Permian volcanic rocks"? I guess you mean that the Brenta drainage area comprises plutonic and volcanic rocks, which are not present in the neighboring Astico and Piave valleys, and that this difference makes it possible to identify the respective deposits? Please make all of this clearer by writing 2-3 more sentences.

AC: We modified the text to avoid misinterpretation and added some sentences, as suggested. Lines 11-14 remains in the same chapter but have been modified to better

reflect our approach.

RC2: p. 5 line 2: "0.0625-2" Did you use this particular grain size fraction in order to compare your dataset with the river endmembers from Garzanti et al. (2006)? If so, please state this in the methods. By taking such a wide grain size window one risks to introduce bias by grain size sorting...

AC: Yes, we used the entire sand fraction in the Gazzi-Dickinson method in order to compare our samples to the endmembers from Garzanti et al. (2006) and Monegato et al. (2010). We modified also the text to make it clear.

RC2: p. 5 lines 4-5: Please rephrase the sentence about the point counting, for example "Following the Gazzi-Dickinson method 400 points per thin section were counted using a 0.5 mm grid spacing (Ingersoll et al. 1984)." Did you use the same grain classes as Vezzoli & Garzanti (2009)? If so, please state so in this paragraph. p. 5 line 6 "Data and parameters were reported in Table 2 AND plotted in ternary diagrams." p. 5 lines 6-9: Please describe more thoroughly the strategy behind this approach (defining endmembers, applying a linear mixing model, reporting the goodness of fit, ....) p. 5 line 12: Avoid the word "adopt" in this context. p. 7 line 4: left side? Please use geographic directions (west/east) p. 7 line 27: "found at THE surface" p. 7 line 28: "Evidence" has no plural: "All evidence shows..." p. 7 line 29: "...which was collecting material from an area located at least 25 km to the north..." p. 8 line 10: right side? Please use geographic directions (west/east) p. 9 line 1: "The lower unit could be attributed to..." p. 9 line 19: What is "CE"? I couldn't find it in the text. p. 9 line 22: "...while the content of felsic volcanic fragments remains high." p. 9 line 23-24: Please rephrase to something like "Although the spectrum of lithic fragments contained in petrofacies 2 is similar to that of petrofacies 1, petrofacies 2 contains more carbonate clasts, generally above 35%. Micritic limestone fragments are particularly common." p. 9 lines 25-28. This is a long and convoluted sentence. Rephrase to something like "The single sample of petrofacies 3 shows a completely different composition. The carbonate fragment content increases to 55% at the expense of quartz (only 10 %) and other grain

types (below 10 %)" ...Also I do not understand how the chert is embedded into the limestone... is it a partially dissolved and then recrystallized limestone or ...? Please specify by including a better petrographic description. p. 10 line 2: "...with an enrichment of carbonate rock fragments..." p. 10 lines 5-6: "Finally, petrofacies 3 with its high carbonate clast content is more similar to the modern Piave River sediment."

AC: Done.

RC2: p. 10 lines 6-9: Couldn't this be interpreted simply as a reworking of deposits from the Piave catchment?

AC: We rephrased the sentences after statistical analysis. The Piave sediments are excluded as the possible source.

RC2: p. 10 lines 13-25: Do "unit 1" and "unit 2" refer to "petrofacies 1" and "petrofacies 2" from before? If so, please use the same name, either "petrofacies" or "unit".

AC: As above, we decided to avoid completely the use of the term "unit" in favor of "petrofacies".

RC2: p. 11 lines 7-8: No capital letters for quartz and feldspar! p. 11 line 10..." and two of them have been dated" p. 11 line 14: Again, no capital letters for quartz and feldspar. p. 11 line 21: Again, no capital letters for quartz and feldspar. p. 11 lines 22-23: "... the dolomite content is significantly higher (Fig. 8)." p. 12 lines 2-3: Again, no capital letters for quartz and feldspar. p. 12 line 28: parts p. 13 line 2: left valley side? Please use geographic directions (west/east) p. 13 line 8: "Based" instead of "Basing"

AC: Done.

RC2: p. 13 lines 8-11: I do not understand this interpretation... Which of your data supports this? Please specify

AC: Referee number 1 had many comments on this section, please refer to its reply for

details.

RC2: p. 13 line 23: consisted of p. 13 line 33: the "so" in the sentence can be deleted p. 15 line 13: Not sure what you mean by "looked to"? Linked to? p. 16 line 1: in respect to

AC: Done.

C) References RC2: I did not thoroughly check all the references, but there are at least two where author's names are not capitalized (e.g. page 19 line 23 "Andò" or page 20, line 1 "Anderson").

AC: We checked and correct references.

D) Comments on figures and tables

RC2: Figure 1: In the legend you use UPPER Permian for the sandstones but EARLY Permian for the plutonic rocks. Use either "Upper and Lower" or "Early and Late" to make this consistent. See for instance Haile 1987 (Marine and Petroleum Geology) for the use of this nomenclature. Figure 2: Increase the size of the yellow square indicating the drill site. Figure 6: What is the red square in the uppermost left corner and why is it red? Figure 7: Add to the figure caption the explanations of the ternary corners (CE, Lm, Lv, Ls...). Did you group together certain grain classes? Table 2: Add a heading for the last column of this table (e.g. "Relative contribution of endmembers" or something like that). Do the numbers (0.886 and so on) refer to R2? Please explain this! What's up with sample RB1-8? Why is it "not representative"?

AC: all comments have been considered and included.

Please also note the supplement to this comment:
https://www.earth-surf-dynam-discuss.net/esurf-2018-22/esurf-2018-22-AC1-supplement.zip

**ESurfD**

---

## Referee Report (RR1)

**Manuscript esurf-2018-22**

**"Glacial dynamics in Prealpine narrow valleys during the Last Glacial Maximum inferred by lowlands fluvial record (NE Italy)"**

**by Rossato et al.**

**Comments on the submitted revised manuscript:**

I checked the submitted revised version against my comments to the initial submission in an attempt to judge how the authors responded to them and whether the manuscript has been improved. The initial submission already was an interesting contribution addressing an interesting and important topic.

My first major comment was addressed in satisfactory fashion and by adding some of the potential scenarios I mentioned. It is follow to follow the arguments by the authors that the lack of terminal moraines is likely caused by glaciofluvial erosion rather than by factors linked to different types of moraine formation. I feel that the manuscript has now been improved regarding this aspect especially for readers not familiar with the specific study area.

With my second major comment regarding the potential blockage of glacier flow by narrow gorges has also been addressed in satisfactory fashion. The authors added some explanations and although I am still not entirely convinced to follow the author's conclusions, I am now totally fine in the way they presented them to the reader and see no reasons to request further changes.

Regarding the technical comments I made to the initial submission, I think the new title is much better than the old one and like the change made. I also acknowledge that the authors took my and the comments of the second reviewer on board and revised the manuscript with regards to the use of commas and other linguistic/stylistic flaws that took my attention when reviewing the initial manuscript. Important information about the type of material radiocarbon-dated has now been added and I would like to thank the authors for providing it. As requested, the difference between lodgement and melt-out till has been clarified in the text.

Summarising, I thanks the authors for addressing my comments on the initial manuscript in satisfactory fashion and now rate the revised manuscript as acceptable for publication.